# Aerosol particle formation in the upper residual layer

**Authors:**

Janne Lampilahti[1], Katri Leino[1], Antti Manninen[2], Pyry Poutanen[1], Anna Franck[1], Maija Peltola[1], Paula Hietala[1], Lisa Beck[1], Lubna Dada[1], Lauriane Quéléver[1], Ronja Öhrnberg[1], Ying Zhou[3], Madeleine Ekblom[1], Ville Vakkari[2,4], Sergej Zilitinkevich[*], Veli-Matti Kerminen[1], Tuukka Petäjä[1,5], Markku Kulmala[1,3,5]

**Affiliations:**

[1]Institute for Atmospheric and Earth System Research / Physics, Faculty of Science, University of Helsinki, Helsinki, Finland.

[2]Finnish Meteorological Institute, Helsinki, Finland.

[3]Aerosol and Haze Laboratory, Beijing Advanced Innovation Center for Soft Matter Science and Engineering, Beijing University of Chemical Technology, Beijing, China.

[4]Atmospheric Chemistry Research Group, Chemical Resource Beneficiation, North-West University, Potchefstroom, South Africa.

[5]Joint International Research Laboratory of Atmospheric and Earth System Sciences, Nanjing University, Nanjing, China.

Correspondence to: Janne Lampilahti (janne.lampilahti@helsinki.fi)

**Abstract:** According to current estimates, atmospheric new particle formation (NPF) produces a large fraction of aerosol particles and cloud condensation nuclei in the earth's atmosphere, therefore having implications for health and climate. Despite recent advances, atmospheric NPF is still insufficiently understood in the lower troposphere, especially above the mixed layer (ML). This paper presents new results from co-located airborne and ground-based measurements in a boreal forest environment, showing that many NPF events (~42%) appear to start in the topmost part of the RL. The freshly formed particles may be entrained into the growing mixed layer (ML) where they continue to grow in size, similar to the aerosol particles formed within the ML. The results suggest that in the boreal forest environment, NPF in the upper RL has an important contribution to the aerosol load in the BL.

## 1. Introduction

It has been estimated that atmospheric new particle formation (NPF) is responsible for most of the cloud condensation nuclei (CCN) in the atmosphere (Dunne et al., 2016; Gordon et al., 2017; Pierce and Adams, 2009; Yu and Luo, 2009). Aerosol-cloud interactions, in turn, have important but poorly-understood effects on climate (Boucher et al., 2013). Being a major source of ultrafine aerosol particles in many environments (e.g. Brines et al., 2015; Posner and Pandis, 2015; Salma et al., 2017; Yu et al., 2019), NPF may have implications for human health.

---

[*] Deceased Feb 15, 2021

NPF has been observed in various environments and at various altitudes inside the troposphere. The majority of NPF observations come from ground-based measurements (Kerminen et al., 2018; Kulmala et al., 2004), which can be argued to represent NPF within the mixed layer (ML). ML is a type of atmospheric BL where turbulence uniformly, especially vertically, mixes quantities like aerosol particle concentrations. Measurements from aircrafts show that NPF is also common in the upper free troposphere (FT) (e.g. Clarke and Kapustin, 2002; Takegawa et al., 2014). Entrainment of particles formed in the upper FT was identified as an important source of CCN in the tropical boundary layer (BL) (Wang et al., 2016; Williamson et al., 2019). Measurements from high-altitude research stations also demonstrate that NPF frequently takes place in the FT, in these cases NPF was often observed in BL air that was transported to the higher altitudes (Bianchi et al., 2016; Boulon et al., 2011; Rose et al., 2017; Venzac et al., 2008).

When studying the vertical distribution of NPF in the lower troposphere one has to consider the evolution and dynamics of the BL. Nilsson et al. (2001) found that the onset of turbulent mixing correlated better with the onset of NPF at ground level than with the increase in solar radiation. The authors gave several hypotheses to why this might be. One hypothesis was that NPF starts aloft, either in the RL or in the inversion capping the shallow morning ML. As the turbulent mixing starts, the newly formed particles would be transported down and observed at the ground-level.

Many observations have supported the hypothesis put forward by Nilsson et al. (2001). Größ et al. (2018), Meskhidze et al. (2019) and Stanier et al. (2004) reported positive correlation between the onset of NPF at ground level and the breakup of the morning inversion due to beginning of convective mixing. Chen et al. (2018), Platis et al. (2015) and Siebert et al. (2004) used in situ airborne measurements and observed that NPF started during the morning on the top of a shallow ML capped by a temperature inversion at a few hundred meters above ground. The particles grew to detectable nucleation mode (sub-25 nm) sizes aloft, and when the ML began to grow due to thermally-driven convection, the particles were mixed downwards and observed at the ground-level where they further continued to grow in size. Stratmann et al. (2003) observed newly formed particles inside the RL disconnected from the shallow ML or the inversion that capped it. Furthermore, Wehner et al. (2010) observed that NPF inside the RL was connected to turbulent layers. On the other hand, Junkermann and Hacker (2018) attributed their observations of elevated ultrafine particle layers at few hundred meter altitudes in the RL to flue gas emissions from smokestacks with subsequent chemistry taking place during air mass transport over long distances.

The hypothesis proposed by Nilsson et al. (2001) was based on observations done in Hyytiälä, Finland, which is a rural site surrounded by boreal forests and with very clean air. However, the supporting evidence comes from measurements done in more polluted environments in Central Europe and USA. Airborne measurements done over Hyytiälä have not found NPF on top of the shallow morning ML or within the bulk of the RL, instead the NPF events seem to start within the ML (Boy et al., 2004; Laakso et al., 2007; O'Dowd et al., 2009). This might be because in the more polluted environments there are high enough concentrations of precursor vapors from anthropogenic sources that NPF can be initiated in the morning inversion and/or within the bulk of the RL. Interestingly, though, observations from Hyytiälä using a small instrumented airplane have frequently found nucleation mode particle layers above the ML at a much higher altitude range of ~1500-2800 m above ground and the explanation for these layers is not clear (Leino et al., 2019; Schobesberger et al., 2013; Väänänen et al., 2016). For example Väänänen et al. (2016) found that for the 2013-2014 airborne measurement campaigns 16/36 (~44%) profiles showed an elevated sub-25 nm particle layer.

In this study we used co-located airborne and ground-based measurements to study nanoparticles over a boreal forest in Hyytiälä, Finland. We aimed to characterize the elevated nucleation mode particle layers that were a frequent observation in the previous studies. Specifically we were looking at the following questions: (1) where in terms of atmospheric layers, how often and why do these aerosol particle layer occur, and (2) how they are related to ground-based observations, and what implications this has for data interpretation.

## 2. Materials and methods

### 2.1. Airborne measurements

We used data from airborne measurement campaigns conducted between 2011 and 2018 around Hyytiälä, Finland. Here we focused on data within 40 km radius from Hyytiälä. Figure 1 shows the data availability from these measurements. Most of the flights were carried out during spring and early autumn because that is when NPF events are most common in Hyytiälä. The measurement setups changed slightly over the years. Detailed descriptions of the setups on board can be found in previous studies (Leino et al., 2019; Schobesberger et al., 2013; Väänänen et al., 2016).

The instrumented aircraft was a Cessna 172 operated from the Tampere-Pirkkala airport (ICAO: EFTP). The sample air was collected through an outside inlet into a main sampling line that was inside the aircraft's cabin. The forward movement of the aircraft during flight provided adequate flow rate inside the main sampling line. The flow rate was maintained at 47 lpm by using a manual valve. The instruments drew air from the main sampling line using core sampling inlets. The necessary flow rate to the instruments was provided by pumps. The airspeed was kept at 130 km/h during the measurement flights.

The aerosol instruments on board considered in this study were an ultrafine condensation particle counter (uCPC, TSI, model: 3776), measuring the >3 nm particle number concentration at a 1-s time resolution, a particle size magnifier (PSM, Airmodus, model: A10) operated with a TSI 3010 CPC, measuring the >1.5 nm particle number concentration at a 1-s time resolution, and a custom-built scanning mobility particle sizer (SMPS) with a short Hauke type DMA and a TSI 3010 CPC, measuring the aerosol number size distribution in the size range of 10-400 nm. The time resolution of the SMPS was about 2.2 min. In addition, basic meteorological data (temperature, relative humidity and pressure) and water vapor concentration from Licor Li-840 gas analyzer were used.

Vertically, the measurement profiles extended approximately from 100 m to 3000 m above the ground. This altitude range covered the ML, RL and roughly 1 km of the FT (Figure 2). The measurement flights lasted about 2-3 hours and were flown mostly during the morning (~6:00-10:00 UTC) and afternoon (~11:00-14:00 UTC). Horizontally, the profiles were flown perpendicular to the mean wind in order to avoid the airplane's exhaust fumes.

### 2.2. Ground-based measurements

Comprehensive atmospheric measurements have been done at the SMEAR II station in Hyytiälä (61°50'40" N, 24°17'13" E, 180 m above sea level) since 1996 (Hari and Kulmala, 2005). The landscape around the site is flat and dominated by Scots pine forests, with small farms and lakes scattered nearby. The station represents typical rural background conditions.

We used data from the BAECC (Biogenic Aerosols–Effects on Clouds and Climate) campaign, which took place in Hyytiälä during Feb-Sep, 2014 (Petäjä et al., 2016), to study the relationship between BL evolution and NPF observed at the station. High spectral resolution lidar (HSRL) measurements and meteorological balloon soundings released every 4 hours by the U.S. Department

of Energy ARM mobile facility allowed us to monitor the evolution of the BL (Nikandrova et al.,

2018).

From the HSRL data we looked at the values of backscatter cross section in order to see the

development of the ML during the day. The data were averaged into 30-m altitude bins and 10-min

temporal bins. The ground-based measurements during the BAECC campaign were also

supplemented by aircraft measurements using the instrumented Cessna. In case of missing

soundings, we also looked at the balloon soundings released from Jokioinen ~120 km south-west

from Hyytiälä (WMO: 02963).

The number size distribution of aerosol particles between 3 and 1000 nm was measured at the

station using a differential mobility particle sizer (DMPS, Aalto et al., 2001). A neutral cluster and

air ion spectrometer (NAIS, Airel Ltd., Mirme and Mirme, 2013) measured the number size

distribution of air ions and particles in the size ranges of 0.8-42 nm and 2-42 nm, respectively

(Manninen et al., 2009). The time resolutions of the DMPS and NAIS were 10 min and 4 min,

respectively. The vertical flux of particles >10 nm was measured by the eddy covariance method

from 23 m above ground, which is a couple of meters above the canopy (Buzorius et al., 2000). The

growth rates for aerosol particles were calculated using the log-normal mode fitting method

described in (Kulmala et al., 2012).

Vertical profiles of horizontal and vertical winds were measured with a Halo Photonics Stream Line

scanning Doppler lidar since year 2016. The Halo Photonics Stream Line is a 1.5 µm pulsed

Doppler lidar with a heterodyne detector and 30-m range resolution, and the minimum range of the

instrument is 90 m (Pearson et al., 2009). At Hyytiälä, a vertical stare of 12 beams and integration

time of 40 s per beam is scheduled every 30 min, whereas the other scan types operated during the

30-min measurement cycle were not utilized in this study. The lidar data were corrected for a

background noise artifact (Vakkari et al., 2019). The turbulent kinetic energy (TKE) dissipation rate

was calculated from the vertical stare according to the method by O'Connor et al. (2010) with a

signal-to-noise-ratio threshold of 0.001 applied to the data. Data availability is limited by relatively

low aerosol concentration at Hyytiälä, but TKE dissipation rate can be retrieved on most days up to

the top of the BL.

**3. Results and discussion**

In the airborne measurements we frequently observed a layer of nucleation mode (sub-25 nm) particles above the ML. First we introduce how the phenomenon was observed in the airborne and ground-based measurements using two case studies. Then we show that sub-25 nm particle layers occurred in the topmost part of the RL by studying the average vertical profile of particle number-size distribution and temperature from the airplane. Then we associate the nucleation mode particles in the upper RL to a specific signal in the ground-based measurements and use the observations at the SMEAR II station to gather long-term statistics. All times are reported in UTC.

### 3.1 Case study: May 2, 2017

On May 2, 2017 during the measurement airplane's ascend over Hyytiälä we observed an increased number concentration of 3-10 nm ($N_{3-10}$) and 1.5-3 nm ($N_{1.5-3}$) particles, approximately between 1200 and 2000 m above sea level (asl), in the top parts of the ML (Figure 3A). The lower edge of the aerosol particle layer was observed at 12:24. Within the particle layer the maximum $N_{1.5-3}$ was ~5000 cm$^{-3}$ and $N_{3-10}$ was ~6000 cm$^{-3}$. Below the particle layer $N_{1.5-3}$ and $N_{3-10}$ were ~3000 cm$^{-3}$. Above the layer $N_{3-10}$ dropped to ~200 cm$^{-3}$. This low number concentration indicates that the airplane was measuring above the ML. The $N_{1.5-3}$ dropped to ~2000 cm$^{-3}$ and further down to ~200 cm$^{-3}$ during the descend. The temperature inversion and the drop in water vapor concentration indicate that the height of the ML was approximately 2200 m asl (Figure 3B).

The PSM sometimes had problems with increasing background number concentration (measured with a filter in front of the inlet) during ascends, especially above 2 km. In these cases the background number concentration would increase as the altitude was increased. It is unlikely that on this day the $N_{1.5-3}$ layer was caused by this kind of instrumental problem alone because the number concentration decreased above the layer.

During the descend the airplane entered back into the ML at 12:56 and the $N_{1.5-3}$ and $N_{3-10}$ were increased throughout the ML. The $N_{1.5-3}$ was around 4000 cm$^{-3}$ and $N_{3-10}$ increased from 4000 cm$^{-3}$ to around 8000 cm$^{-3}$ towards the surface. On the same day, an early morning flight before the sunrise was also performed (Figure 3A). During this flight no elevated aerosol particle layer was observed and the number concentrations were quite uniform with altitude in the different size ranges, staying below 1500 cm$^{-3}$.

Roughly 10 min after the aerosol particle layer was first observed from the airplane during the ascend, a new particle mode with similar-sized particles (geometric mean mode diameter about 10 nm) appeared at the ground-level at 12:36 (Figure 3C). This time was estimated from the NAIS measurements. The appearance of this new particle mode was characterized by a negative peak in the vertical particle flux, suggesting that the particles could be mixed down from aloft. The new particle mode continued to grow for several hours despite the airmass moving over Hyytiälä, indicating a large horizontal source area for the particles. At the ground level a new particle mode with lower number concentration coupled with negative particle flux also appeared at around 10:00.

The number concentration of >3 nm aerosol particles along the afternoon flight track is shown in Figure 3D. The particle layer was observed roughly 4 km north of Hyytiälä. Throughout the flight the particle number concentration was higher in the north compared to the south. To take this horizontal variability into account we only included aerosol data from the northern part of the flight track in Figure 3A. The particle layer could still appear in the airborne data and later in the ground-based data if the particles were transported from north to south during the measurement period due to a change in wind direction. Wind measurements from the SMEAR II mast at 67.2 m altitude show that the wind direction changed from 290 degrees to 330 degrees between 12:00-12:30 (Figure 3E). The particles were observed at the SMEAR II station right after the wind direction had changed. On the other hand the negative particle flux associated with the appearance of the particles would suggest and elevated source and in the case of airmass change we would expect to see the particles appear during the change in wind direction, not after it. In any case it is difficult to say conclusively if the aerosol particle observations on this day were due to vertical or horizontal transport.

The airmasses came from the Arctic Ocean over northern Scandinavia. They went over the west coast of Finland where there are known pollution sources, however in Hyytiälä the $SO_2$ and CO levels remained low all day (~0.025 ppb and ~115 ppb for $SO_2$ and CO, respectively). Even when the particles were observed at the measurement station no increase in pollutant concentrations was observed. Pollution released into the night time RL from elevated sources such as flue gas stacks would be expected to form layers at lower altitudes, below few hundred meters. If the pollution is released during daytime, it is expected to be uniformly mixed into the ML and stay like that in the RL (Junkermann and Hacker, 2018).

In order to study the atmospheric layers in the lower troposphere we plotted the TKE dissipation rate calculated from the Doppler lidar measurements during May 1-2, 2017 and temperature soundings from Jokioinen (Figure 3F). In the Doppler lidar measurements, the increase in the TKE dissipation rate reveals the development of the ML on both days. On May 1, 2017 the ML reached roughly 1900 m asl. The temperature sounding at 18:00 shows that this mixed layer was capped by a thermal inversion at about 2000 m asl. In the two subsequent soundings during the night the inversion stayed at roughly the same altitude and marked the top of the RL. In the temperature sounding on May 2, 2017 at 12:00 only one inversion is observed at about 1900 m asl suggesting that at this point the RL was already mixed into the growing ML. The lidar measurement agrees that on May 2, 2017 the ML reached 1900 m asl around 12:00. About 25 min later the aerosol particle layer was observed from the Cessna. These observations are supported by the temperature and water vapor profiles measured on board the Cessna during the morning and afternoon flights (Figure 3B).

### 3.2 Case study: May 19, 2018

On May 19, 2018 a similar case was observed. Figure 4A shows that during the airplane's ascend the lower edge of the particle layer was observed at ~1200 m asl and the top of the layer was at 2000 m asl. The $N_{3-10}$ increased in the layer from ~1000 cm$^{-3}$ up to ~10000 cm$^{-3}$. When the airplane descended back into the ML the $N_{3-10}$ was increased to around 6000 cm$^{-3}$ throughout the ML. The temperature and water vapor measurements show that a well-mixed layer was capped by inversion at 2000 m asl (Figure 4B). Unfortunately the PSM was not working during this flight.

Figure 4C shows that horizontally the particle layer was observed approximately 5 km west of the SMEAR II station. When the airplane entered back into the ML the particle number concentration was increased over the SMEAR II station and in the west part of the measurement area. The aircraft only flew ~2 km east of the SMEAR II station before turning southwest towards the airport, so it is unclear if the number concentration was increased in the east as well. There was no appreciable change in wind direction, which was from the north, during the measurement period (Figure 4D). Therefore it is unlikely that the particles in the layer were horizontally transported to Hyytiälä from west to east.

The air masses arrived from a similar sector as in the May 2, 2017 case (Arctic Ocean over northern Scandinavia). $SO_2$ and CO concentrations in Hyytiälä remained low during the measurements (~0.05 ppb and ~127 ppb for $SO_2$ and CO, respectively).

Figure 4E shows particle number size distribution measurements from the measurement airplane and from the field station. The particle layer was observed as increased number concentrations in the smallest size channels of the SMPS at 9:00 before the airplane flew above the ML. Roughly 20 minutes later a similar-sized particle mode appeared in the ground-based data. For this day there were no particle flux data. The new particle mode continued to grow larger inside the ML for several hours.

Figure 4F shows the TKE dissipation rate on May 18-19, 2018 from Hyytiälä and temperature soundings from Jokioinen. On May 18, 2018 the ML went up to 2500 m asl in Hyytiälä. The Jokioinen soundings show that at 6:00 the top of the RL was at about 1800 m asl, marked by the subsiding inversion left from the previous day's ML. The top of the particle layer was at approximately 2000 m asl.

### 3.4 Evidence of nanoparticles in the upper RL based on long-term airborne measurements

In the two case studies above the aerosol particle layer was associated with the altitude where the top of the RL was. In order to study this connection further we analyzed the airborne data measured during 2011-2018. In Figure 5 we plotted the median and $75^{th}$ percentile number size distributions measured on board the aircraft as a function of altitude during NPF event days (65 days out of 130 measurement days) between 07:00 and 10:00 UTC. This is the time window when the morning measurement flight was usually done. NPF event days are characterized by a new growing particle mode appearing in the sub-25 nm size range (Dal Maso et al., 2005). If aerosol formation in the upper RL occurs on less than half of the NPF event days, it might not be visible in the median plot, but might still appear in the $75^{th}$ percentile plot.

Interestingly, in the $75^{th}$ percentile plot a layer of nucleation mode particles is observed at 2500-3000 m above sea level. This altitude range is well above the still growing ML at 07:00-10:00. We wanted to know if the elevated particle layer was associated with a temperature inversion, since the RL is commonly capped by such an inversion (Stull, 1988). In Figure 5 we plotted the mean temperature profile from the flights when the $N_{10-25}$ in 2000-3000 m altitude range exceeded the $75^{th}$ percentile $N_{10-25}$ value (18 days).

The temperature profile shows an inversion base at 2500 m and this is likely where on average the top of the RL was. The reason for the unusually deep RL is probably that the NPF event days tend to be sunny spring days and the ML can grow exceptionally high, which also leads to a deep RL. Our finding is in line with previous observations by Schobesberger et al. (2013) who measured nucleation mode particles close to an elevated temperature inversion above the ML on multiple measurement flights over southern Finland.

### 3.5 Connection between nanoparticles in the upper RL and ground-based observations

With the BAECC dataset we wanted to investigate whether the sudden appearance of nucleation mode particles with downward particle flux was associated with the ML reaching the upper RL. This would not only further test the hypothesis that the nanoparticles reside the topmost part of the RL, but also provide us with a condition to identify these events from the ground-based data alone.

We looked for cases where a new particle mode suddenly appeared in the nucleation mode size range during the daytime and the first observation of these particles was associated with a negative peak in particle flux. We noted the times when the particles first appeared, and also estimated a confidence interval of the observation. Then we checked if we could find out the height of the RL from balloon soundings or the Cessna flights. We looked for an elevated temperature inversion that was roughly at the same altitude as the previous day's maximum ML height, which was determined from HSRL and/or sounding. We noted the base height of the temperature inversion and took this as the top of the RL. Then we followed the height of the new ML from the HSRL measurements and noted the time when the ML reached the inversion base, also estimating a confidence interval. Figure 6 illustrates an example for this procedure.

We found 8 cases during the campaign where the analysis could be fully carried out and they are summarized in Table 1. Figure 7 shows a positive correlation between the new particle mode appearance time and the time when the ML reached the top of the RL. This suggests that the suddenly appearing nucleation mode particles were entrained into the ML from the upper RL. We found only a weak positive correlation between the new particle mode appearance time and the geometric mean diameter of particles in the new mode at the moment they were first observed. The mean growth rate of the appearing particle modes was 2.2 nm h$^{-1}$ which is similar to 2.5 nm h$^{-1}$ reported by Nieminen et al. (2014) for 3-25 nm particles during NPF events in Hyytiälä.

The time that the ML reaches the upper RL depends on the height of the RL, which in turn depends on the height of the ML on the previous day and the rate at which the top of the RL subsides. The mixing time also depends on the rate at which the ML on the day of interest grows. For example on March 28, 2014 the ML height on the previous day and the RL height during the night were 1300 m and 1100 m, respectively. On April 4, 2014 the corresponding numbers were 2800 m and 2200 m. Because of this on March 28, 2014 the ML reached the upper RL much earlier at ~7:00 compared to April 4, 2014 when the ML reached the upper RL at ~11:00. For example on April 15, 2014 the ML grew slowly in the morning due to presence of low clouds that limited convection. Because of this the ML reached the top of the RL relatively late at 13:00.

In a well-mixed layer we would expect the entrained particles to reach the surface in less than an hour (Stull, 1988). If the BL was stratified the particles could reach the surface at very different rates which might significantly distort the results in Figure 7. The balloon soundings indicate that the MLs in the 8 cases were well-mixed since the potential temperature profiles calculated from soundings released around noon and late afternoon were almost constant up to the top of the ML (see example profile in Figure 6).

### *3.6 Proposed explanation for the results*

One possible explanation for the elevated nucleation mode particle layers could be long-range transport coupled with changes in the particle number size distribution such as particle shrinkage. However, it is not clear why such process would favor the RL-FT interface. If the particle emissions were released into the ML they would likely be distributed more or less uniformly throughout the RL and not be concentrated at the top of the RL. If the transported particles subsided from the FT, we would expect to see particle layers at various altitudes in the FT on different days, and the layers would not be localized at the top of the RL. We studied the origin of the airmasses in the particle layers and found that they were mostly coming from the so-called "clean sector" in the northwest of Hyytiälä (Figure 8). During other than winter months this sector is associated with non-polluted air and NPF from natural precursors (Tunved et al., 2006).

We find the most likely explanation to be NPF in the upper RL. The gaseous precursors involved in NPF may end up in the upper RL because of mixing from the surface during the previous day (e.g. organic vapors emitted from the forest or sulfuric acid, ammonia and amines originating from human activities) or because of long-range transport in the FT (e.g. iodine oxides from the ocean).

377

Many factors favor NPF at higher altitudes, including enhanced photochemistry, reduced sinks and reduced temperature. However, the NPF inducing features of the upper RL would probably be linked to the mixing that takes place in the interface between the RL and FT, since this is the place where the particle layers seem to be limited to. Nilsson and Kulmala, (1998) found that mixing two air parcels with different initial temperatures and precursor vapor concentrations can lead to a considerable increase in the nucleation rate. Therefore mixing air from the RL and FT over the inversion, where the precursors are present in one of the layers (most likely the RL), could induce aerosol particle formation in the interface layer.

Another possibility is that the RL and the FT contain different precursor vapors that did not initiate nucleation or particle growth on their own, however when the vapors are mixed in the interface between the two layers NPF occurs. For example on May 2, 2017 Beck et al. (in preparation) measured the composition of naturally charged ions using a mass spectrometer on board an aircraft concurrently with our measurements. It was found that during the first flight (~02:30-04:00 UTC) the chemical composition was different in the RL compared to the FT. For example highly oxygenated molecules (HOMs) as well as iodine containing compounds were present in the RL while methanosulfonic acid (MSA) and sulfuric acid were detected in the FT.

If the growing ML reaches the upper RL, the newly formed particles will be mixed downwards into the ML where they continue to grow in size as low-volatility vapors present in the ML are able to condense onto these particles. The processes are illustrated in Figure 9. In case the particles will not be mixed down, they may persist in the FT for a longer time period and possibly have stronger contribution to cloud formation.

### 3.7 Implications for classifying NPF events

Previous studies that classified NPF events observed in Hyytiälä have collected statistics on the occurrence of suddenly appearing particle modes. Buenrostro Mazon et al. (2009) classified the so-called undefined days between 1996-2006 from Hyytiälä. The undefined days are days that do not fit the NPF event or the nonevent day classes (Dal Maso et al., 2005). One category the authors used was "tail events" where a new particle mode appears at particle diameters greater than 10 nm and grows for several hours. The authors found that 26% of NPF events were tail events (assuming that tail events were also NPF events). Dada et al. (2018) collected statistics on "transported events"

where elevated number concentration of 7-25 nm particles persisted for more than 1.5 hours, but no elevated number concentrations at smaller particle sizes were observed. It was found that 36% of the NPF events observed for over 10 years in Hyytiälä were "transported events". They occurred especially when the conditions inside the ML were less favorable for nucleation.

Here we found cases in the SMEAR II data between 2013 and 2017, in which a new growing particle mode suddenly, without continuous growth from smallest detectable sizes (3 nm), appears in the nucleation mode (sub-25 nm) and is associated with a negative peak in the vertical particle flux (upper RL NPF). We also noted cases where a new particle mode appears with a continuous growth from the smallest detectable sizes (ML NPF). Based on the previous analysis we assume that in the former case NPF took place in the upper RL and in the latter case inside the ML. The analysis included 1750 days.

The monthly fractions of the different cases are shown in Figure 8. We found that NPF within the ML occurred on 13% (234/1750) of all the days and NPF in the upper RL on 7% (117/1750) of all the days. During spring (Mar-May) the corresponding percentages were 31% (132/431) and 17% (74/431). On many days NPF took place both in the upper RL and within the ML (4% or 74/1750 of all days and 12% or 53/431 of spring days). According to this analysis, NPF in the upper RL constitutes 42% (117/277) of the NPF event days in Hyytiälä.

The monthly distribution of upper RL NPF events follows the distribution of ML NPF events, with a peak during spring (Mar-May). This is well in line with previous studies that classified NPF events in Hyytiälä (Dal Maso et al., 2005; Nieminen et al., 2014). This makes sense since the conditions favoring ML NPF would also favor upper RL NPF. However, Buenrostro Mazon et al. (2009) and Dada et al. (2018) found that the tail events and transported events had a peak during the summer months (Jun-Aug).

On 16% of the NPF event days NPF only took place in the upper RL but not in the ML. This number is smaller than the 36% found by Dada et al. (2018) for transported events and the 26% found by Buenrostro Mazon et al. (2009) for tail events. This might be because we restricted to cases where a negative peak in particle flux was associated with the appearance of nucleation mode particles. For example, a case where the particles were horizontally advected to the measurement site would not be expected to cause a negative peak in the particle flux and therefore would not be classified as upper RL NPF.

445

### *4. Conclusions*

446

447

We measured aerosol particles, trace gases and meteorological parameters on board an instrumented Cessna 172 over a boreal forest in Hyytiälä, Finland. The airborne data was complemented by the continuous, comprehensive ground-based measurements at the SMEAR II station.

451

We found multiple evidence of nanoparticle layers situated in the topmost part of the RL. Many points would suggest that the particle layers originated from NPF in the upper RL: the particles were in the sub-25 nm size range, the airmasses originated from a sector north-west of Hyytiälä that is associated with NPF and less pollution during non-winter months (Tunved et al., 2006), ground-based observations show continuous growth over several hours indicating a large horizontal source area instead of a point source and increased nucleation rate would be expected to occur in the inversion between RL and FT (Nilsson and Kulmala, 1998). We estimate that such upper RL NPF occurs on 42% of the NPF event days in Hyytiälä. Our results provide new information on NPF in the BL and they should be taken into account when interpreting and analyzing ground-based as well as airborne measurements of aerosol particles.

462

**Data availability:** The particle flux and DMPS data can be accessed from https://avaa.tdata.fi/web/smart/smear  (Junninen et al., 2009; last access: Oct 1, 2020). The BAECC HSRL and radiosonde data is available from https://adc.arm.gov/discovery/ (Bambha et al., 2014; Keeler et al., 2014); last access: Oct 1, 2020). The Jokioinen soundings can be accessed using the Finnish Meteorological Institute's open data service https://en.ilmatieteenlaitos.fi/open-data (last access: Oct 1, 2020). The ERA5 dataset can be accessed from https://cds.climate.copernicus.eu/cdsapp#!/home (last access: May 6, 2020). The rest of the data was gathered into a dataset that can be accessed from https://zenodo.org/record/4063662#.X3cHQnUzY88 (Lampilahti et al., 2020; last access: Oct 2, 2020).

472

**Author contribution:** JL, KL, AM, PP, AF, MP, PH, LD and LQ conducted the airborne measurements in 2017. PP wrote processing script for the airborne data. RÖ classified the SMEAR II data for NPF events between 2013-2017. LB, SZ, VMK, TP and MK contributed to the data analysis. YZ and ME analyzed the airborne data between 2011-2018. VV provided the Doppler lidar data. JL prepared the manuscript with contributions from all co-authors.

478

**Acknowledgements:** This project has received funding from the ERC advanced grant No. 742206, the European Union's Horizon 2020 research and innovation program under grant agreement No. 654109, the Academy of Finland Center of Excellence project No. 272041 and from the Academy of Finland grant 314 798/799. We thank Erkki Järvinen and the pilots at Airspark Oy for operating the research airplane and we are grateful for their hospitality and helpfulness.

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

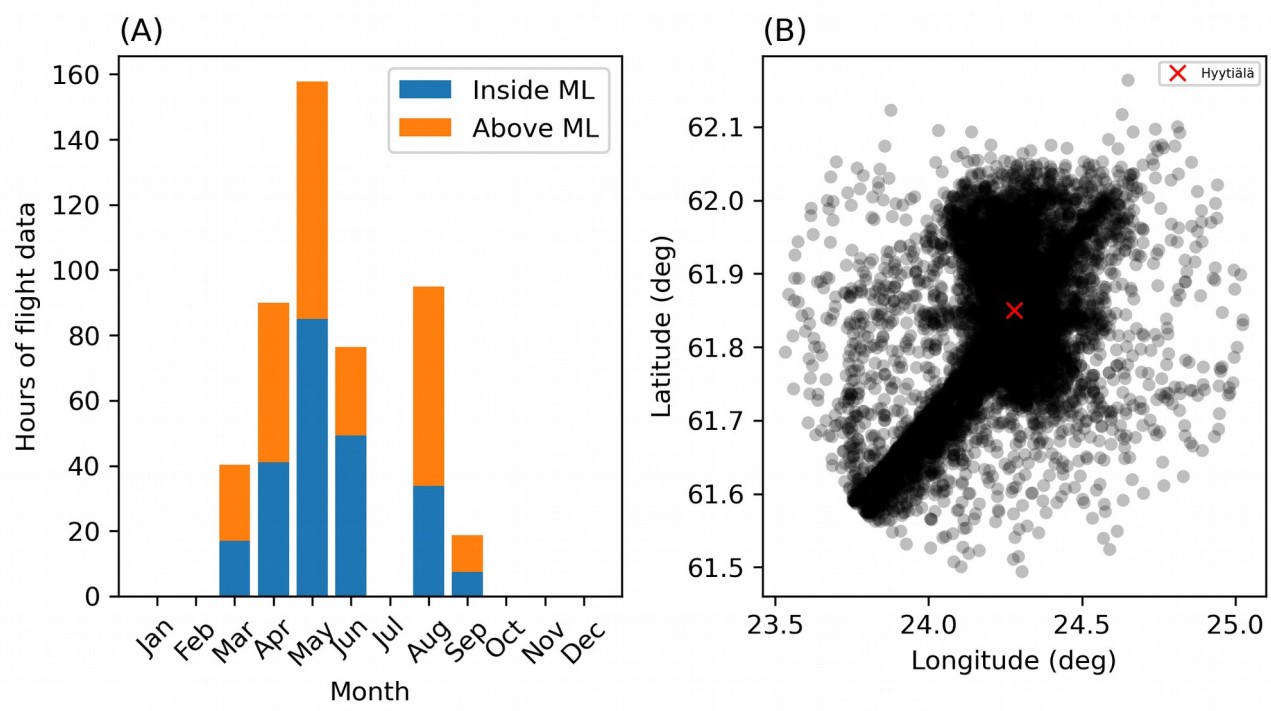

Figure 1: (A) monthly airborne data availability between 2011-2018 divided into measurements above and below the ML, based on the ML height obtained from the ERA5 reanalysis data. (B) horizontal distribution of the 2011-2018 airborne measurement data. We chose the data within 40 km radius from Hyytiälä.

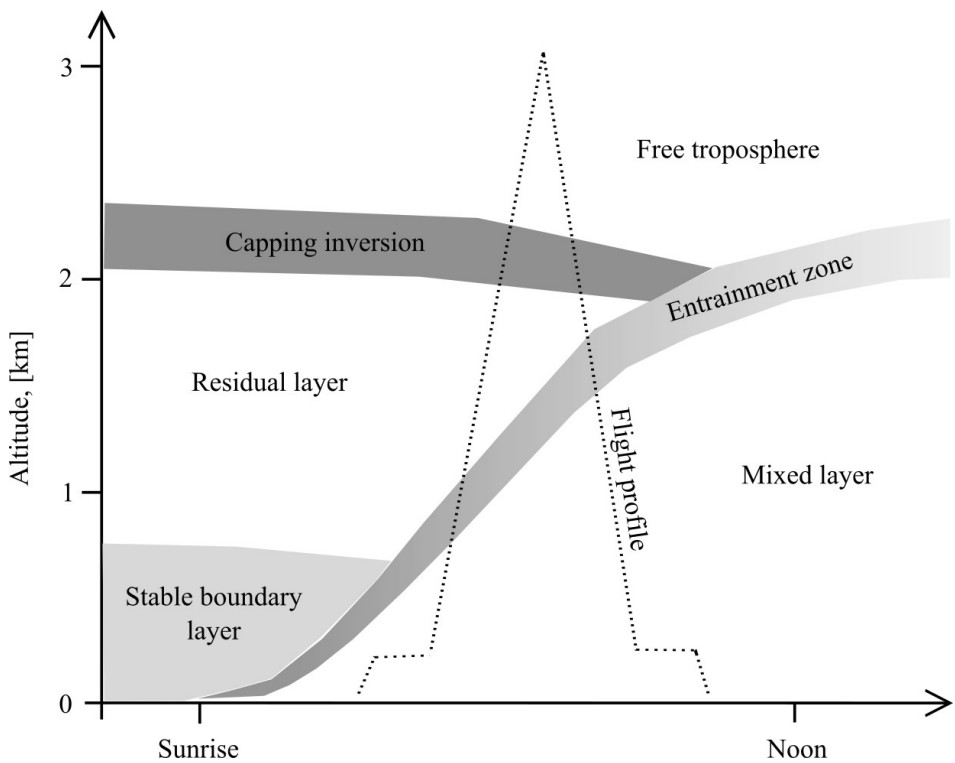

Figure 2: A schematic diagram of an average flight profile in relation to BL evolution.

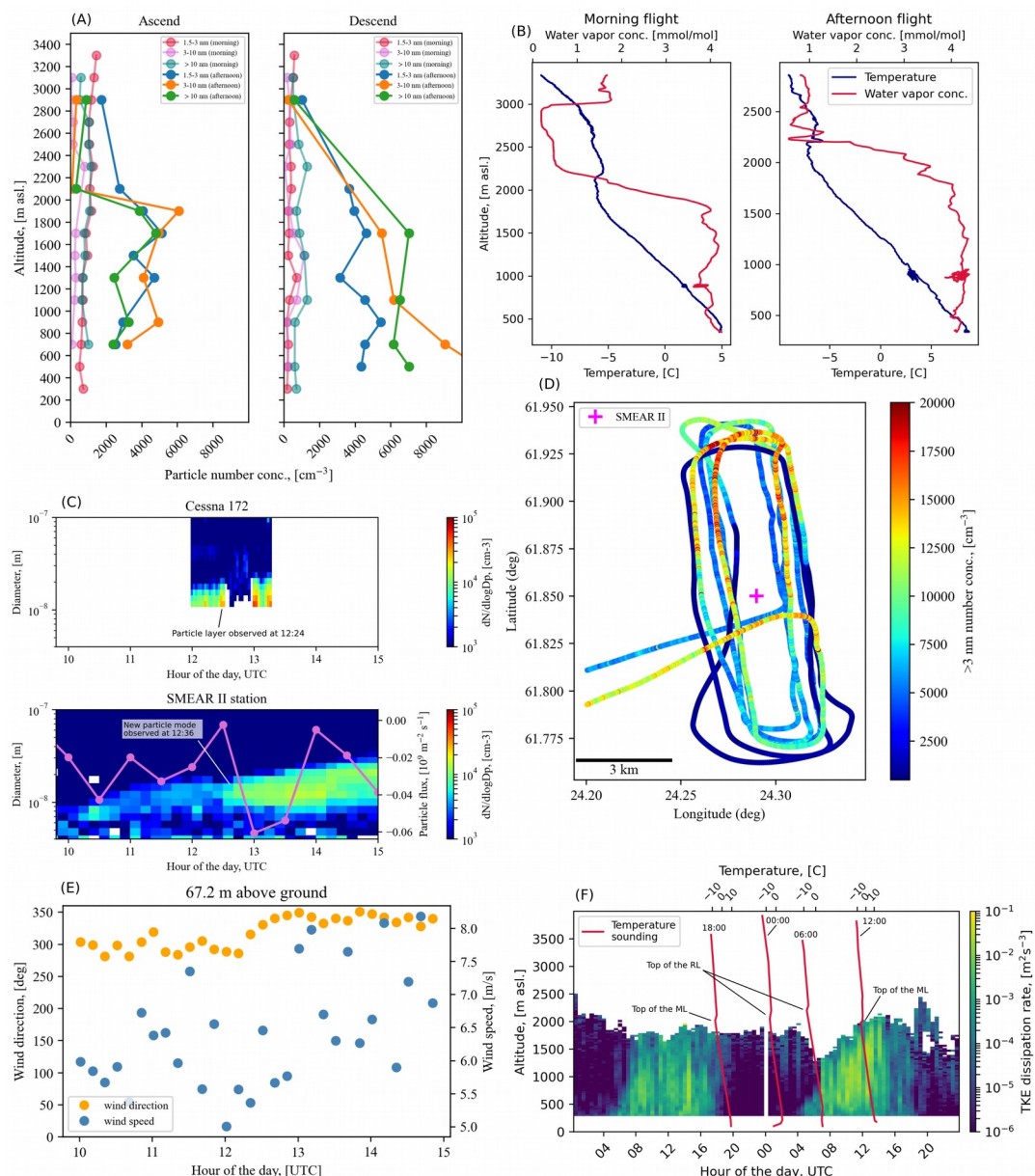

Figure 3: Panel (A) shows vertical profiles of aerosol particle number concentration in three different size ranges (1.5-3 nm, 3-10 nm and >10 nm) on May 2, 2017. The data shows the morning filght (02:26-03:55 UTC) and the afternoon flight (12:00-13:20 UTC). The profile from the afternoon flight is restricted to the northern part of the flight track (E:24.25-24.35, N:61.875-61.95). Panel (B) shows the temperature and water vapor concentration profiles from the morning and the afternoon ascends. Panel (C) shows the particle number-size distribution from the measurement airplane and the SMEAR II station. The vertical flux of >10 nm particles is superimposed. Negative means downward and positive upward particle flux. Panel (D) shows the afternoon filght track colored by >3 nm particle number concentration. Panel (E) shows the wind speed and direction from the SMEAR II mast (67.2 m). Panel (F) shows turbulent kinetic energy (TKE) dissipation rate measured by the Doppler lidar in Hyytiälä between May 1-2, 2017. Temperature soundings from Jokioinen are superimposed.

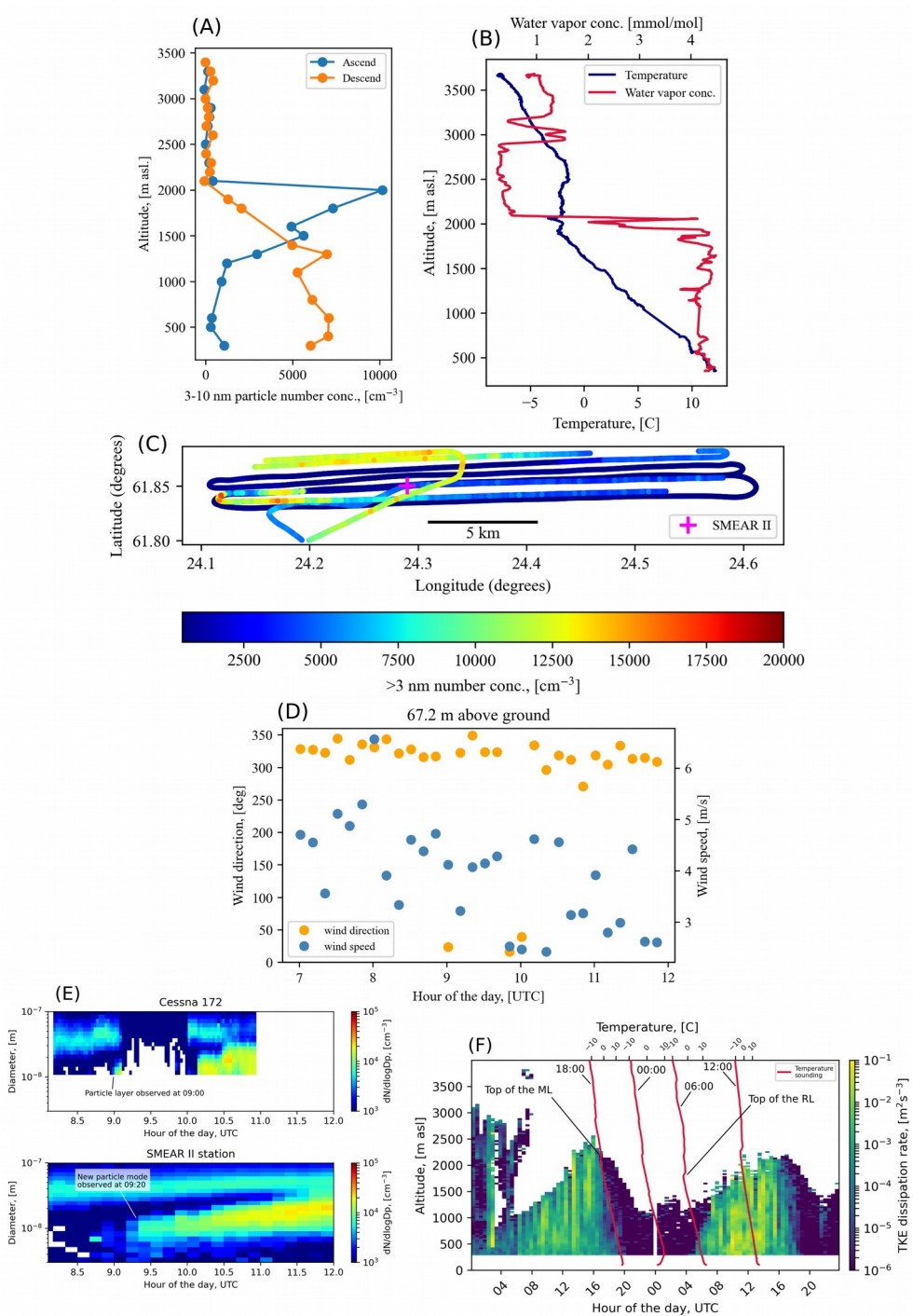

Figure 4: Panel (A) shows vertical profiles of 3-10 nm particle number concentration on May 19, 2017 between 8:42-10:24 UTC. Panel (B) shows the temperature and water vapor concentration profiles druing the ascend. Panel (C) shows the afternoon filght track colored by >3 nm particle number concentration. Panel (D) shows the wind direction and speed measured from the SMEAR II mast at 67.2 m. Panel (E) shows the particle number-size distribution from the measurement airplane and the SMEAR II station. Panel (F) shows turbulent kinetic energy (TKE) dissipation rate measured by the Doppler lidar in Hyytiälä between May 18-19, 2018. Temperature soundings from Jokioinen are superimposed.

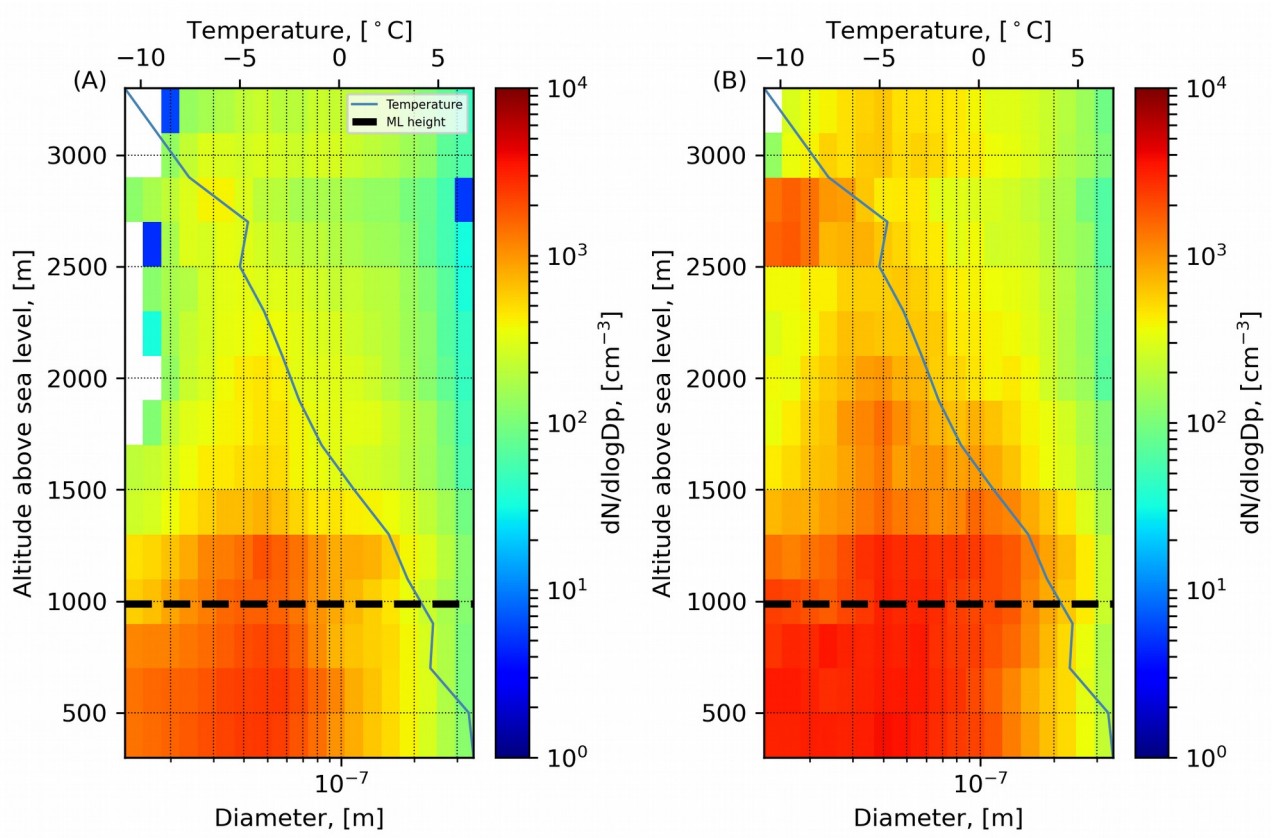

Figure 5: Panel (A) shows the median and panel (B) the 75th percentile vertical profile of particle number-size distribution measured on board the Cessna on NPF event days between 9-12 AM. The number-size distribution was binned into 200 m altitude bins. The data is from the campaigns conducted between 2011-2018. The dashed line is the mean ML height obtained from the ERA5 reanalysis data. The blue line shows the mean temperature profile from measurment flights when the sub-25 nm number concentration in the 2000-3000 m altitude range was above the 75th percentile.

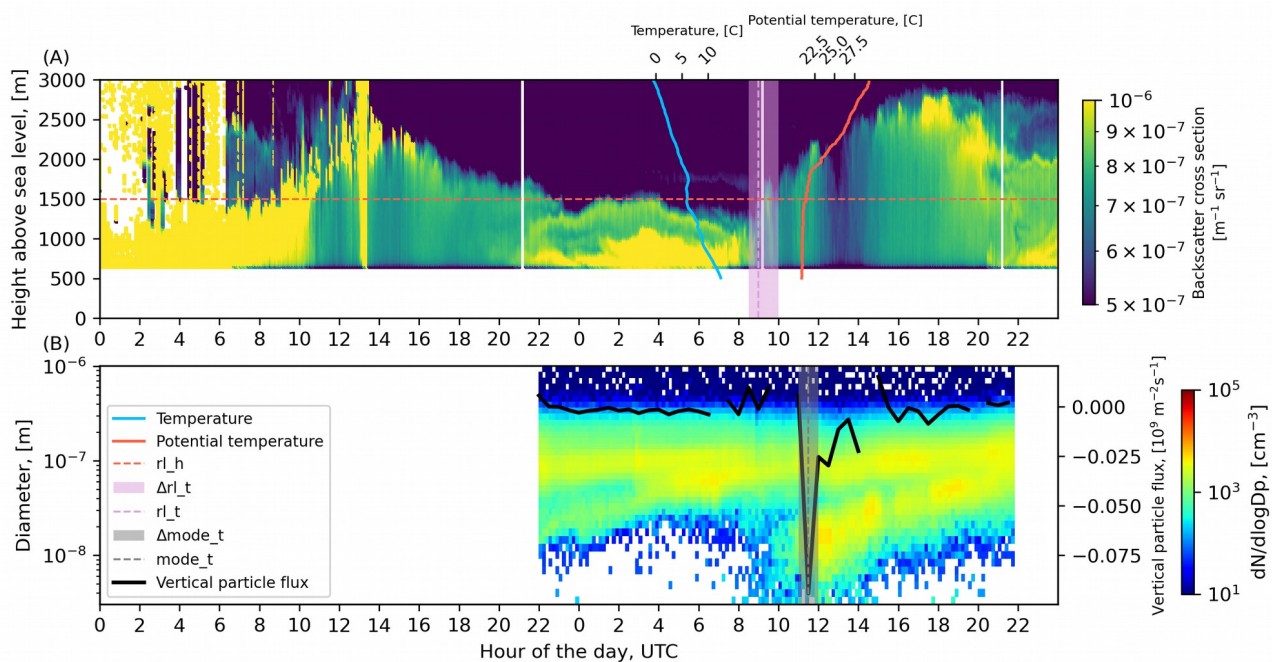

Figure 6: Panel (A) shows the backscatter cross section measured by the HSRL on June 4-5, 2014. The development of the ML is visible from the backscatter cross section signal. Temperature and ptoential temperature form soundings released in Hyytiälä at 5:20 and 11:20 on June 5, 2014 respectively are superimposed. The horizontal line rl_h refers to the height of the inversion base in the sounding (height of the RL). The rl_t and Δrl_t refer to the time when the ML was estimated to reach the rl_h and the confidence interval for this time, respectively. Panel (B) shows the particle number-size distribution measured at the SMEAR II station, the black line is the vertical particle flux. The mode_t and Δmode_t respectively refer to the time and the confidence interval, when a nucleation particle mode that is associated with downward particle flux suddenly appears.

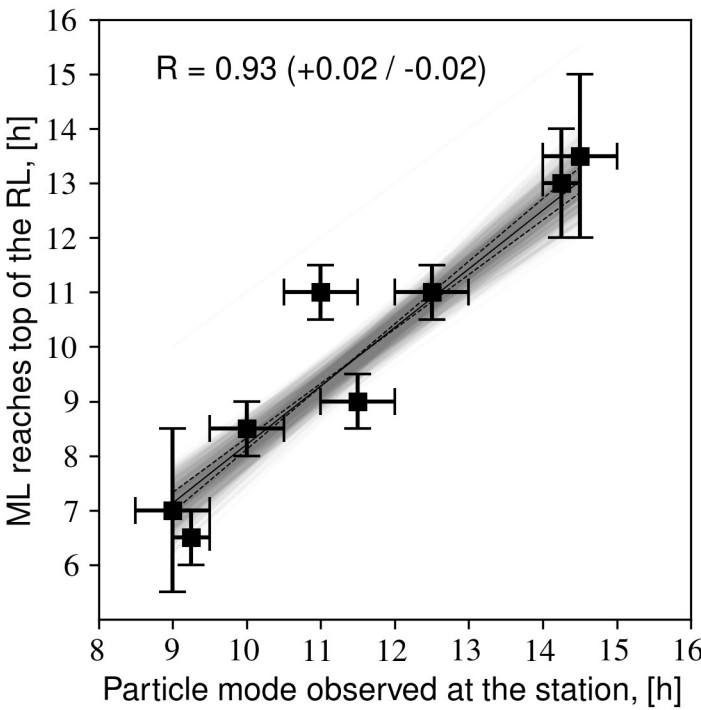

Figure 7: The correlation between the times when a new particle mode coupled with downward particle flux was observed at the field site and the times when the ML reached the top of the RL.

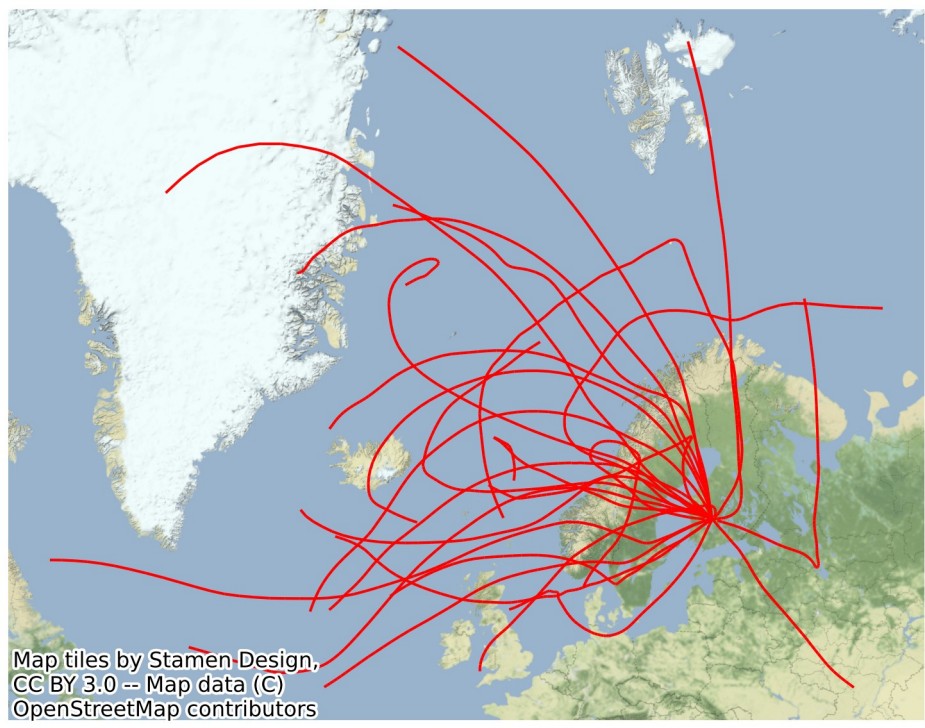

Figure 8: Airmass back trajectories arriving to altitude over Hyytiälä where nucleation mode particle layers were located based on airborne data and the BAECC data. We calculated the airmass histories for 72 hours, however in the figure some of the trajectories are truncated to fit the map. The trajectories were calculated based on two different conditions. First, based on the BAECC data analysis (Section 3.5) such that the airmass arrived at the top of the RL when the ML reached that altitude (see Table 1 for these altitudes and times). Second, based on the vertical profiles between 2011-2018 (Section 3.4) such that the back trajectories arrived at 2600 m altitude at 10:00 UTC on the days when the $N_{10-25}$ in 2000-3000 m altitude range exceeded the $75^{th}$ percentile $N_{10-25}$ value.

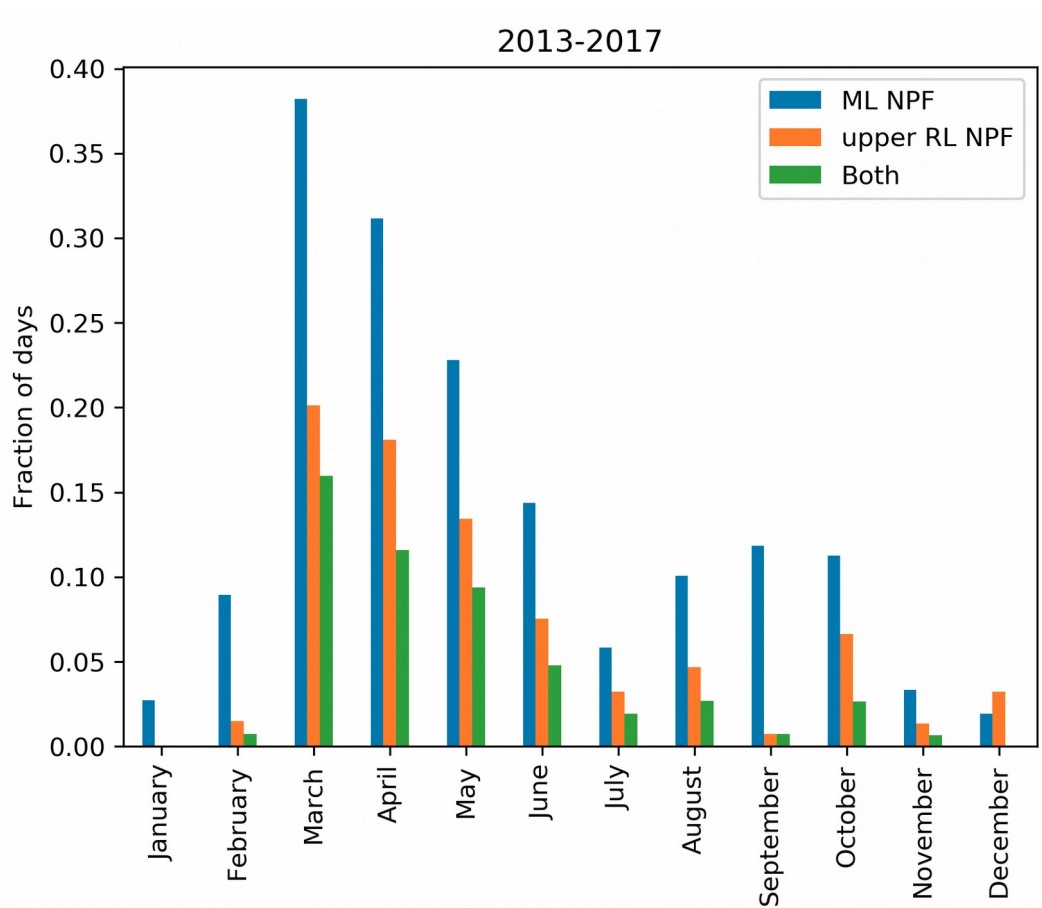

Figure 9: Monthly fractions of NPF within the ML and NPF in the upper RL in Hyytiälä between 2013-2017.

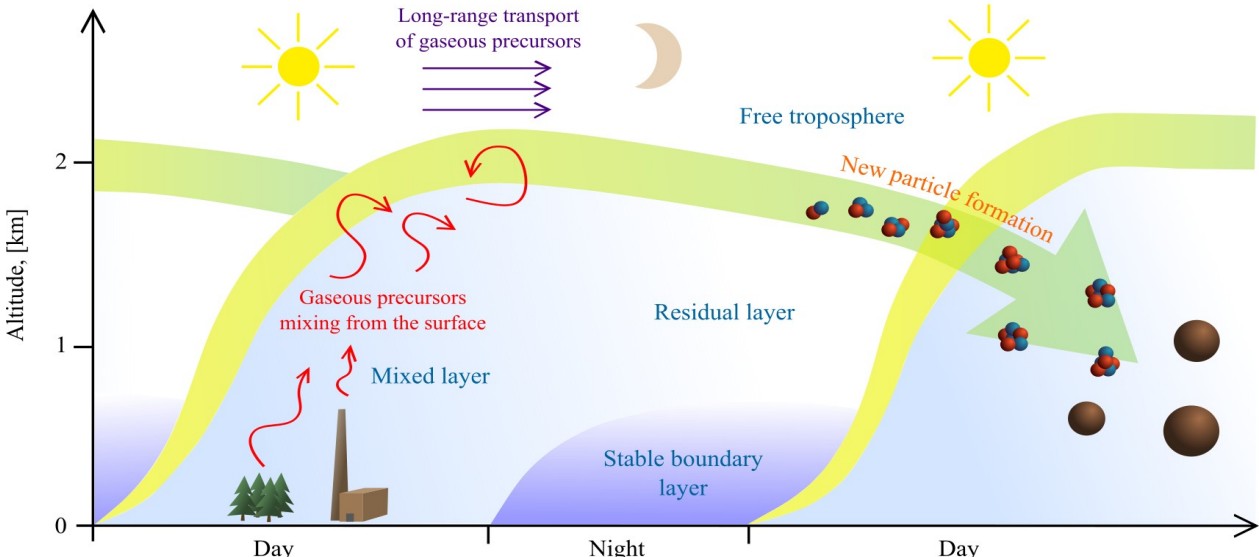

Figure 10: Schematic drawing illustrating the proposed mechanism behind NPF in the upper RL. Gaseous precursors released from biogenic and/or anthropogenic sources are mixed throughout the ML. When the mixing stops during the night the gases are stuck in the RL. Also gaseous precursors may be transported in the FT. In the following morning photochemistry begins and aerosol particles are formed in the interface between the RL and the FT. The freshly formed particles remain in the elevated layer or get mixed into the a new ML if it reaches the height of the upper RL. The aerosol particles continue to grow larger, contributing to the aerosol load in the BL.

Table 1: rl_h = residual layer height during night or early morning (m asl), rl_ht = time when the rl_h was observed (time when the sounding was released, hour of the day, UTC), mode_t = nucleation mode particle mode first appears (hour of the day, UTC), mode_t1/mode_t2 = nucleation mode particle mode appearance confidence interval (hour of the day, UTC), rl_t = new mixed layer reaches the top of the residual layer (hour of the day, UTC), rl_t1/rl_t2 = new mixed layer reaches the top of the residual layer confidence interval (hour of the day, UTC), bl_h = observed maximum height of the previous day's boundary layer (m asl.), dp = mean mode diameter for the newly appeared particle mode, when they first appear (nm), gr = growth rate calculated for the newly appeared partice mode (nm h$^{-1}$), pf = the value of the negative particle flux peak ($10^9$ m$^{-2}$ s$^{-1}$).

| date | rl_ht | rl_h | mode_t1 | mode_t | mode_t2 | rl_t1 | rl_t | rl_t2 | dp | bl_h | pf | gr |
|---|---|---|---|---|---|---|---|---|---|---|---|---|
| 20140328 | 5.3 | 1100 | 8.5 | 9 | 9.5 | 5.5 | 7 | 8 | 20 | 1300 | -0.25 | 2.28 |
| 20140331 | 7.6 | 2400 | 14 | 14.5 | 15 | 12 | 13.5 | 14 | 10 | 2200 | -0.06 | 2.1 |
| 20140404 | 8.5 | 2200 | 10.5 | 11 | 11.5 | 10.5 | 11 | 11.5 | 8 | 2800 | -0.04 | 1.39 |
| 20140409 | 5.5 | 1500 | 9 | 9.25 | 9.5 | 6 | 6.5 | 7 | 8 | 1800 | -0.13 | 1.18 |
| 20140415 | 5.3 | 1600 | 14.5 | 14.25 | 15 | 12 | 13 | 14 | 11 | 1700 | -0.18 | 1.94 |
| 20140422 | 0.0 | 1800 | 12 | 12.5 | 13 | 10.5 | 11 | 11.5 | 17 | 1900 | -0.17 | 1.0 |
| 20140518 | 0.0 | 1500 | 9.5 | 10 | 10.5 | 8 | 8.5 | 9 | 13 | 1900 | -0.11 | 2.91 |
| 20140705 | 5.3 | 1500 | 11 | 11.5 | 12 | 8.5 | 9 | 10 | 12 | 1700 | -0.1 | 4.83 |

488

