# Peer review of "Aerosol particle formation in the upper residual layer"

_Atmospheric Chemistry and Physics, 2020_

## Referee Comment (RC1) · Anonymous Referee #1 · 2 Nov 2020

The manuscript is well written and easy to follow. There are two main drawbacks to this paper. 1) it is lacking the substance. More case studies, with detected start and end times and particle growth rates, are needed. 2) the paper does not acknowledge previous studies on the topic and brings little new scientific insight into the topic of residual layer nucleation events. For example, the discussion regarding the residual layer nucleation particle growth after the entrainment in the mixed layer is limited to a schematic drawing. However, I still think this paper might be of scientific interest. I, therefore, would recommend accepting this paper for publication after the authors have addressed the following issues.

Introduction

A number of previous studies, i.e., Nilsson et al. (2001), Stratmann et al. (2003),
Stanier et al. (2004); (Wehner et al., 2007), and (Platis et al., 2016) suggested that enhanced turbulent mixing, related to the growth of daytime convective boundary layer and the lift of the inversion could cause downward mixing of the particles, which had already grown in size. In addition, there have been several recent studies that point out direct evidence for NPF occurring aloft, in the interface between the shallow convection and inversion (Chen et al., 2018; Größ et al., 2018). By using turbulence statistics and the boundary layer dynamics (Meskhidze et al., 2019) and (Zimmerman et al., 2020) quantified the frequency of the residual layer and the ground level nucleation events and assessed their contributions (relative to other sources) to the near-surface fine particle number budgets during different seasons. The authors don't seem to acknowledge many of these studies. That leaves the impression that the residual layer nucleation and the particle entrainment into the mixed layer is a novel mechanism for explaining the appearance of >10 nm-sized particles at the near-surface layer. I would encourage the authors to clearly discuss how their research builds upon these prior studies and highlight the similarities.

Results and discussion

The airplane flight profiles seem to be different between Fig. 3 and Fig. 4. Are these two different profiles? If so, please explain.

Fig. 4 shows that the negative flux was measured at the surface starting at 9:30 am. However, according to Fig. 3, there was no significant vertical gradient between the surface and the 1000 m. Please explain the presence of strongly negative fluxes between 9:30 am and 12:30 pm. According to Fig. 4, a new 10 nm particle mode only appeared at the ground-level at ~12:35 pm. So, what causes negative fluxes in the morning?

Please include several more case studies so the reader can compare the similarities and contrast the differences. For each case study please show the normalized spectral density plots so the reader can ascertain that there was indeed a growth event following

the appearance of >10 nm-sized particles at the near-surface layer.

Please include the flux values for each of the 8 cases shown in Fig. 8. Since the DMPS was running at the ground site, it would be interesting to know the detected start and the end time of the events, as well as the growth rate for different size particles.

Fig. 8 shows 6-hour differences between the times when the mixed layer reaches the top of the residual layer. Please provide an explanation based on the full analysis of the meteorological data.

Please compare the monthly fractions of new particle formation events (Fig. 9) in Hyytiälä with the data reported in other studies discussed above.

References

Chen, H., Hodshire, A. L., Ortega, J., Greenberg, J., McMurry, P. H., Carlton, A. G., Pierce, J. R., Hanson, D. R. and Smith, J. N.: Vertically resolved concentration and liquid water content of atmospheric nanoparticles at the US DOE Southern Great Plains site, Atmospheric Chemistry and Physics, 18(1), 311–326, doi:10.5194/acp-18-311-2018, 2018.

Größ, J., Hamed, A., Sonntag, A., Spindler, G., Manninen, H. E., Nieminen, T., Kulmala, M., Hõrrak, U., Plass-Dülmer, C., Wiedensohler, A. and Birmili, W.: Atmospheric new particle formation at the research station Melpitz, Germany: connection with gaseous precursors and meteorological parameters, Atmospheric Chemistry and Physics, 18(3), 1835–1861, doi:10.5194/acp-18-1835-2018, 2018.

Meskhidze, N., Jaimes-Correa, J. C., Petters, M. D., Royalty, T. M., Phillips, B. N., Zimmerman, A. and Reed, R.: Possible Wintertime Sources of Fine Particles in an Urban Environment, Journal of Geophysical Research: Atmospheres, 124(23), 13055–13070, doi:10.1029/2019JD031367, 2019.

Nilsson, E. D., Rannik, U., Kulmala, M., Buzorius, G. and O'Dowd, C. D.: Effects of continental boundary layer evolution, convection, turbulence and entrainment, on aerosol

formation, Tellus B, 53(4), 441–461, doi:10.1034/j.1600-0889.2001.530409.x, 2001.

Platis, A., Altstädter, B., Wehner, B., Wildmann, N., Lampert, A., Hermann, M., Birmili, W. and Bange, J.: An Observational Case Study on the Influence of Atmospheric Boundary-Layer Dynamics on New Particle Formation, Boundary-Layer Meteorology, 158(1), 67–92, doi:10.1007/s10546-015-0084-y, 2016.

Stanier, C. O., Khlystov, A. Y. and Pandis, S. N.: Nucleation Events During the Pittsburgh Air Quality Study: Description and Relation to Key Meteorological, Gas Phase, and Aerosol Parameters Special Issue of Aerosol Science and Technology on Findings from the Fine Particulate Matter Supersites Program, Aerosol Science and Technology, 38(sup1), 253–264, doi:10.1080/02786820390229570, 2004.

Stratmann, F., Siebert, H., Spindler, G., Wehner, B., Althausen, D., Heintzenberg, J., Hellmuth, O., Rinke, R., Schmieder, U., Seidel, C., Tuch, T., Uhrner, U., Wiedensohler, A., Wandinger, U., Wendisch, M., Schell, D. and Stohl, A.: New-particle formation events in a continental boundary layer: first results from the SATURN experiment, Atmos. Chem. Phys., 15, 2003.

Wehner, B., Siebert, H., Stratmann, F., Tuch, T., Wiedensohler, A., PetäJä, T., Dal Maso, M. and Kulmala, M.: Horizontal homogeneity and vertical extent of new particle formation events, Tellus B: Chemical and Physical Meteorology, 59(3), 362–371, doi:10.1111/j.1600-0889.2007.00260.x, 2007.

Zimmerman, A., Petters, M. D. and Meskhidze, N.: Observations of new particle formation, modal growth rates, and direct emissions of sub-10 nm particles in an urban environment, Atmospheric Environment, 117835, doi:10.1016/j.atmosenv.2020.117835, 2020.
* * *

---

## Referee Comment (RC2) · Wolfgang Junkermann (Referee) · 17 Nov 2020

The manuscript describes aircraft based vertical profiles of nanometer sized particles in the range from 1.5 to 400 nm with a main emphasis on the lower particles sizes of 1.5 to 3 nm and 3-20 nm. The authors summarize data from 7 years of flight experiments and conclude that a larger fraction of ground based nanoparticle events (NPE's) (> 40% of observations) occurs in cases when small particles are transported downwards from the residual layer into the planetary boundary layer. A case study is shown as well for one day in May, 2017 to explain the interpretations. A schematic drawing of the proposed processes is included.

General comments

[Figure]

Although the paper is easy to read it's obvious that there is a significant lack of data supporting the conclusions in the manuscript, especially information about the history of the air mass under investigation and potential pollution. The title is misleading. The main message of the title, that this study is an investigation of aerosol formation is not justified. For such an investigation the CESSNA could be a suitable mobile platform but, completely different flight patterns and an upgraded instrumentation would be necessary (see below).

There is a second message behind, not mentioned in the manuscript title: Nanoparticles appearing at the surface as new particles are actually produced elsewhere and advected towards the ground under sunny conditions by a diurnal cycle of vertical transport. This is well I agreement with the observation that solar radiation is a major driving parameter behind the appearance of new nanoparticles at the surface (Baranizadeh et al, 2014). The authors now allocate the place where the particles are formed into the residual layer. However, they neglect that also in the residual layer a horizontal transport process is running, on a larger scale and less, but not completely independent of diurnal patterns. The argument, that particles might be produced elsewhere holds as well for this location / altitude range. Now, where are these particles produced?

The authors claim that nanoparticle formation appears to start in the residual layer, a statement that is not confirmed and that the processes linking transport and nanoparticle appearance are not well understood disregarding a century of atmospheric physics research. They do not discuss the production pathways, the contributing chemical precursors and their sources and the already available literature about gas to particle conversion in the atmosphere (e.g. Mohnen and Lodge, 1969, Gillani et al, 1978).

To investigate the origin of atmospheric nanoparticles, a sound 3D-meteorological analysis is mandatory. In the current manuscript such an analysis is missing. A few vertical profiles of temperature, potential temperature and water vapor are included. For these data scales and units are missing. Horizontal transport, wind (speed and direction), although mentioned in Fig 10 of the manuscript as part of the proposed mechanism for

new particle formation, is not taken into consideration at all. Previous studies on the production and transport of ultrafine particles are not adequately taken into account.

While vertical mixing can be a fast process with a time constant of less than an hour (Georgii, 1956) (a better estimate from the current investigation?) aerosol formation and growth is a process of several hours (Kulmala et al, 2013). Accordingly and in agreement with fig. 10 (from the manuscript) the history of the air mass at least for several hours need to be considered for particles in the size range < 20 nm. It's not important where the air mass is originated on the long term, it's more important, whether and how it is polluted on the way to the observation location and by which chemical or particulate compounds.

In the case study or May 2, 2017, relatively clean, regionally unpolluted air in the morning is compared to a polluted case after noon. The authors assumption of a clean arctic air masses (line 166) for all profiles during the day, a prerequisite for the conclusion that aerosols suddenly appear from gas to particle conversion is thus not valid. Thus, neither a local nor a spatial atmospheric aerosol formation can be derived from these early morning and afternoon profiles.

Fig. 1 shows the results of a HYSPLIT backtrajectory analysis for the may 2 case study and upwind pollution sources with elevated emission altitude ($\sim$ 200 – 400 m). Note the windsheer. The white circle is the 20 km radius range around Hyytiälä, the dotted line the 40 km radius. MBL at 05 UTC was at 260 m over Vaskiluoto, the main source in the area. The sulphur emission rate is $\sim$ 150 kg/h since installation of the desulfuration equipment in 1993 (http://www.energico.fi/ref-vaskiluoto.shtml), compare to 250 kg/h from the power station Karlsruhe and its primary nanoparticle emission (Junkermann et al, 2011a). Paper mills and smelters are similar sources for particulate emissions (Ayers et al, 1979, Bai et al, 1992, Rosenfeld, 2000, Junkermann et al 2011a, Brachert et al, 2013, see also the Finnish national emission inventory, https://www.eea.europa.eu/data-and-maps/dashboards/air-pollutant-emissions-data-viewer-3). Ammonia as a neutralizing compound is widely

used in the fossil fuel industry to suppress NO2 emissions and it is available in high concentrations internally (primary emissions) and also externally as ammonia slip (Li et al, 2017).

The main result of the investigation of the current manuscript is the analysis, that a significant fraction of Hyytiälä nanoparticle events are correlated to enhanced nanoparticle concentrations in the residual layer and are caused by downward transport of this nanoparticle aerosol, driven by thermal convection.

The statements in the conclusions about evidence for NPF and unique thremodynamic conditions (not shown) in line 276/278 about upper residual layer aerosol formation again are speculative and not supported by data. Although only occasionally reported (Kerminen et al, 2018 and literature cited therein) physically also shrinking of aerosols cannot be excluded. The fate of nanoparticles during transport depends on ambient conditions as well as on the presence of other aerosols (coagulation, condensation sink). This information is missing. The statement, that meteorology, but mandatory in 3D and including wind, has to be taken into account for interpretation and analyzing of ground based and airborne data (lines 279 to 281) and the statement that the current results are important for NPF events elsewhere in the world however, can be fully supported.

Questions:

What are the ambient conditions relevant to particle formation or aging in the residual layer, temperature, humidity, water vapor concentration, wind speed and direction, potential condensation sink? Are aging nano-particles in this layer growing or shrinking (Kerminen et al, 2018 and references cited therein)?

40 km is a wide range, see Fig. 1. Where is the GPS-location of the measurements with respect to well known locations of major precursor molecule and/or primary nanoparticle emissions upwind?

What is the flight pattern during ascents and descents? Can this be used to derive wind data from example from GPS when the Lidar is not sensitive enough?

Why are the measurements in the ascend beginning at 200 m, the descend ends at 600 m agl? Teisko, $\sim$ 15 km (alt 158 m) from Hyytiälä (alt 178 m) would be a location for missed approaches and legal low altitude flying. O'Dowd et al (2009) presented profiles nearly to the ground from QUEST 2003.

Are there any indicators for example from the Hyytiälä meteorological tower that can be related to vertical mixing intensity? Potential parameters could be surface temperature or temperatures in the vertical profile. Glider pilots use ground based temperature measurements for a decision when to take off.

Minor comments

Although an SMPS is onboard there is no size distribution presented for the case study. A complete size distribution would be a mandatory information for the interpretation as it carries information about the age of the particles (and potential distance to the source location).

For comparison of airborne and ground based data the same parameters, particle size distributions and not particles on the aircraft and air ions on the ground, should be used.

Whether the vertical profiles within $2\frac{1}{2}$ hours in the early afternoon and another flight in the morning are comparable at all remains open, see the HYSPLIT trajectories above. The vertical profiles of the morning flight including air mass history and trajectory need to be included as well.

To investigate, whether the 1.5 nm particles grow into the size range of 3-20 nm and to disentangle between NPF in a sulphur rich environment and primary emissions a better size resolution is necessary (Junkermann et al, 2011a). See there and in subsequent papers size distributions with a finer resolution in the range below 10 nm.

It needs a lagrangian flight pattern to confirm that airmass change is excluded, see Junkermann and Hacker (2015).

The observations in the 3-20 nm range are well in agreement with the patchy structure of particle number and size distributions from previous studies in the area (O'Dowd et al, 2009, Schobesberger et al, 2013, Väänänen, et al, 2016, Leino et al, 2019) as well as the patchwork blanket of power station plumes shown by Junkermann et al (2016). All these studies point towards a significant contribution from local emission hotspots. Chemical analysis from 20 years of particle research at Hyytiälä reveals that sulphur molecules and likely ammonia are among the key substances required for the production of nanoparticle clusters. A recent publication by Hao et al (2018) about measurements at Hyytiälä even requires particulate sulphate in the residual layer mixed downward to explain the observations on the ground.

The sources of such sulfate particles in the atmosphere are well known and typically linked to burning processes (Bigg and Turvey, 1978, Ayers et al, 1979, Whitby et al, 1978). In Finland these sources are mostly located along the coastline, about 200 km or approximately 5-6 hours upwind of Hyytiälä, (Fig. 1, www.endcoal.org). Further examples for primary nanoparticle size, aging and emission rates are shown in the papers of Junkermann et al.

Transport via the residual layer is not an exclusive pattern, veering plumes from wind direction changes in the planetary boundary layer can explain as well the observations without additional aerosols in the residual layer (examples: O'Dowd et al, 2009, for Hyytiälä under conditions with snow covered ground or Laaksonen et al, 2005 (SPC, Italy) Junkermann and Hacker (2018)). In all cases 3D-meteorology is the key for analysis of these observations.

Final comments:

Recent and historic literature is not always taken into account. There are not many airborne studies of nanoparticles, but they should be included.

Time within the manuscript is mixed between UTC and Eastern European Summer Time (EEST) in figures and text.

Fig. 5: Scales for Theta and water vapor are missing. The figure is not really supportive, it suggests a high mixed layer at night although the upper rim of the Lidar data reflect only the vertical range of the measurements. Significant TKE for vertical mixing is restricted only to daylight hours.

Line 166, please take into account: a few hours upwind of Hyytiälä one of Finland's largest pollution source ~150 kg sulphur dioxide / his located, emitting both a large amount of primary particles and a mixture of substances relevant to nanoparticle formation independent on the time of the day.

Fig. 9 should be discussed in terms of the annual variability of meteorology, for example the intensity of convection under typical weather conditions in Finland. The intensity of the vertical mixing process described in the manuscript is dependent on surface conditions (snow until the end of March?) and surface and vertical profile temperatures throughout the year.

Fig. 10 is outdated and needs severe revision. Sources are not always on the ground they can be elevated as well.

Platis et al 2015 should be Platis et al, 2016

Junkermann and Hacker (2018) is cited in the text but missing in the reference list.

References

Ayers, G.P., Bigg, E.K., and Turvey, D.E. 1979: Aitken Particle and Cloud Condensation Nucleus Fluxes in the Plume from an Isolated Industrial Source. J. Appl. Met., 187, 449-459

Bai, H., Biswas, P. and Keener, T. C., 1992, "Particle Formation by NH3-SO2 Reactions at Trace Water Conditions,"Ind. Eng. Chem. Res., 31, 88.

[Figure]

Baranizadeh, E., Arola, A., Hamed, A., Nieminen, T., Mikkonen, S., Virtanen, A., Kulmala, M., Lehtinen, K., Laaksonen, A., 2014, The effect of cloudiness on new-particle formation: investigation of radiation levels, Boreal Environmental Research, 19 (suppl. B): 343–354

Bigg, E.K., Ayers, G.P. and Turvey, D.E. ,1978, Measurement of the dispersion of a smoke plume at large distances from the source, Atmos. Environ., 12, 1815-1818

Bigg, E.K., and Turvey, D.E., 1978, Sources of atmospheric particles over Australia, Atmos. Environ., 12, 1643-1655

Brachert, L., T. Kochenburger, and K. Schaber, 2013, Facing the sulfuric acid aerosol problem in flue gas cleaning: Pilot plant experiments and simulation. Aerosol Sci. Technol., 47, 1083–1091, https://doi.org/10.1080/02786826.2013.824549.

Georgii, H.W., 1956: Flugmeteorologie, Akademische Verlagsgesellschaft, Frankfurt, Germany, 2nd edition, in German

Gillani, N.V., Husar, R.B., Husar, J.D., Patterson D.E., and Wilson, W.E., 1978, Project MISTT, Kinetics of particulate sulphur formation in a power plant plume out to 300 km, Atmospheric Environment, 12, 589-598

Hao, L., Garmash, O., Ehn, M., Miettinen, P., Massoli, P., Mikkonen, S., Jokinen, T., Roldin, P., Aalto, P., Yli-Juuti, T., Joutsensaari, J., Petäjä, T., Kulmala, M., Lehtinen, K. E. J., Worsnop, D. R., and Virtanen, A., 2018, Combined effects of boundary layer dynamics and atmospheric chemistry on aerosol composition during new particle formation periods, Atmos. Chem. Phys., 18, 17705–17716, https://doi.org/10.5194/acp-18-17705-2018.

Junkermann, W., Hagemann, R. and Vogel, B. 2011(a): Nucleation in the Karlsruhe plume during the COPS / TRACKS - Lagrange experiment, Q. J. Roy. Met. Soc., 137, 267-274

Junkermann, W., Vogel, B. and Sutton, M.A. 2011(b): The climate penalty for clean

fossil fuel combustion, Atmos. Chem. Phys., 11, 12917-12924

Junkermann, W., and Hacker, J.M., 2015, Ultrafine particles over Eastern Australia: an airborne survey, Tellus B, 67, 25308, http://dx.doi.org/10.3402/tellusb.v67.25308

Junkermann, W., Vogel, B. and Bangert, M., 2016, Ultrafine particles over Germany - an aerial survey, Tellus B, 68, 29250, http://dx.doi.org/10.3402/tellusb.v68.29250

Junkermann W., and Hacker J., 2018, Ultrafine particles in the lower troposphere: major sources, invisible plumes and meteorological transport processes, BAMS, 99, 2587-2622, Dec. 2018, DOI:10.1175/BAMS-D-18-0075.1

Kerminen, V.-M., Chen, X., Vakkari, V., Petäjä, T., Kulmala, M. and Bianchi, F.: Atmospheric new particle formation and growth: review of field observations, Environ. Res. Lett., 13(10), 103003, doi:10.1088/1748-9326/aadf3c, 2018.

Kulmala, M. , Kontkanen, J., Junninen, H., Lehtipalo, K., Manninen, H.E., Nieminen, T., Petäjä, T., Sipilä, M., Schobesberger, S., Rantala, P., Franchin, A., Jokinen, T., Järvinen, E., Äijälä, M., Kangasluoma, J., Hakala, J., Aalto, P.P., Paasonen, P., Mikkilä, J., Vanhanen, J., Aalto, J., Hakola, H., Makkonen, U., Ruuskanen, T., Mauldin 3rd R.L., Duplissy, J., Vehkamäki, H., Bäck, J., Kortelainen, A., Riipinen, I., Kurtén, T., Johnston, M.V., Smith, J.N., Ehn, M., Mentel, T.F., Lehtinen, K.E.J., Laaksonen, A., Kerminen, V-M. and Worsnop, D.R., 2013, Direct observations of atmospheric aerosol nucleation, Science, 22;339(6122):943-6. doi: 10.1126/science.1227385.

Leino, K., Lampilahti, J., Poutanen, P., Väänänen, R., Manninen, A., Buenrostro Mazon, S., Dada, L., Franck, A., Wimmer, D., Aalto, P. P., Ahonen, L. R., Enroth, J., Kangasluoma, J., Keronen, P., Korhonen, F., Laakso, H., Matilainen, T., Siivola, E., Manninen, H. E., Lehtipalo, K., Kerminen, V.-M., Petäjä, T. and Kulmala, M., 2019, Vertical profiles of sub-3 nm particles over the boreal forest, Atmospheric Chem. Phys., 19(6), 4127–4138, doi:10.5194/acp-19-4127-2019.

Li Z., Jiang, J., Ma, Z., Fajardo, O.A., Deng, J., and Duan, L., 2017, Influence of flue

gas desulfurization (FGD) installations on emission characteristics of PM2.5 from coal-fired power plants equipped with selective catalytic reduction (SCR), Environ. Poll., 230, 655-662. doi: 10.1016/j.envpol.2017.06.103

Mohnen, V. A. and Lodge, J.P., 1969, General review and survey of Gas-to-Particle Conversions, Proceedings of the 7th International Conference on Condensation and Ice Nuclei. Podzimek, J. (ed.). Prague Academia, Publishing House of the Czechoslovak Academy of Sciences (1969) 69-91

O'Dowd, C. D., Yoon, Y. J., Junkermann, W., Aalto, P., Kulmala, M., Lihavainen, H., and Viisanen, Y.: Airborne measurements of nucleation mode particles II: boreal forest nucleation events, Atmos. Chem. Phys., 9, 937–944, https://doi.org/10.5194/acp-9-937-2009, 2009.

Platis, A., Altstädter, B., Wehner, B. et al. An Observational Case Study on the Influence of Atmospheric Boundary-Layer Dynamics on New Particle Formation. Boundary-Layer Meteorol 158, 67–92 (2016). https://doi.org/10.1007/s10546-015-0084-y

Quan, J., Liu, Y., Liu, Q., Jia, X., Li, X., Gao, Y., Ding, D., Li, J. and Wang, Z.: Anthropogenic pollution elevates the peak height of new particle formation from planetary boundary layer to lower free troposphere, Geophys. Res. Lett., 44(14), 7537–7543, doi:10.1002/2017GL074553, 2017

Rosenfeld, D., Suppression of rain and snow by urban and industrial air pollution, Science,287, 1793-1796, 2000.

Schobesberger, S., Väänänen, R., Leino, K., Virkkula, A., Backman, J., Pohja, T., Siivola, E., Franchin, A., Mikkilä, J., Paramonov, M., Aalto, P. P., Krejci, R., Petäjä, T. and Kulmala, M.: Airborne measurements over the boreal forest of southern Finland during new particle formation events in 2009 and 2010, Boreal Environ. Res., 18(2), 145–164, 2013.

Väänänen, R., Krejci, R., Manninen, H. E., Manninen, A., Lampilahti, J., Buenrostro

Mazon, S., Nieminen, T., Yli-Juuti, T., Kontkanen, J., Asmi, A., Aalto, P. P., Keronen, P., Pohja, T., O'Connor, E., Kerminen, V.-M., Petäjä, T., and Kulmala, M.: Vertical and horizontal variation of aerosol number size distribution in the boreal environment, Atmos. Chem. Phys. Discuss., https://doi.org/10.5194/acp-2016-556

Whitby K.T., Cantrell B.K., Kittelson D.B., 1978: Nuclei formation rates in a coal-fired power plant plume. Atmos. Environ. 12, 313–321.

**[ACPD](ACPD)**
[Figure]

[Figure]

**Fig. 1.** Fig. 1 HYSPLIT analysis (GDAS 0.5) 8 h backtrajectories for Hyytiälä at 06, 09, 12, 15 UTC, at 800 (red) and 1800 m (yellow) and upwind pollution sources

---

## Referee Comment (RC3) · Anonymous Referee #3 · 22 Nov 2020

This paper describes the influence of dynamics and especially the presence of freshly emitted particles in the residual layer on new particle events observed at the surface. As reported by referee 1, the manuscript is well written and straight forward. However and as reported by referee 1, many references are missed and the quality of the paper could be improved by adding more substance to it.

Major comments

- Roughly isokinetic sampling : Could you please be more precise. The inlet is either isokinetic with a control of the flow within the inlet or not isokinetic. It seems that you are controlling it with a valve and with a constant speed of the Cessna. Therefore most of the time the sampling should be isokinetic. However, roughly is too vague. What are the deviation from the isokinetic conditions ? This condition has a large impact on

[Figure]

the measurement quality and therefore on their validity. Please correct and add more information about that.

- Figure 3 analysis : "The layer had increased number concentrations of sub-20nm and sub-3nm particles." in comparison o what ? The descent profile ? I think you should clearly name the reference you are comparing these results to. Moreover, you should definitely show the profiles from the early morning flight on Figure 3. That would raise no doubts that the aerosol layer was not present before the sun rise and that could give the reader a clear reference. "at this point there were no signs of the particle layer" This is misleading. The layer didn't disappear spread into lower layers, in this case the ML. Is there a threshold for the RL height ? I believe the highest is the better due to lower temperature and cleaner air. But is there any RL height range for those events ? Could you also add the ML height in this figure ?

- L 168-171 : The NPF starts at 12:36 but the vertical particle flux show minimum values at 10:30 et 13:00. If aerosols are coming from the residual layer (around 1700m), the process is not instantaneous right ? So the NPF should be related to the minimums of Vertical particle flux occurring at 10:30 and 11:30. Can you estimate the vertical speed of the aerosols ? Is the aerosol speed playing a role in the NPF occurrence ? I would think that yes due to the fact that slow motion aerosol would have grown to much larger sizes ? Could you run the analysis also for non event days ? Is there a vertical wind speed threshold that need to be exceeded ? Also for other NPF cases linked to RL NPF events, Can you tell us more about the vertical particle flux patterns observed before the occurrence of NPF ? Is it different for each case ?

- Figure 6 : I'm not sure what you plotted on this figure. The color code correspond to dN/dlogDp (cm-3). So is it a total concentration or is it from a specific bin ? It must be a specific bin and most probably within the fine diameter range due to the conclusions drowned. Could you please provide the percent of NPF event linked to aerosol formation in the upper layer ? Then you could used this result to justify the 75th percentile use.

Minor remarks

L52 : need to define ML

L147 : In the aircraft data : not well said

Figure 7 : Need to be more precise : - early morning of June 5th : 0 – 4h ? Is there a reason why you choose that time to determine the Residual layer ? could you provide some stat for each cases of the delay between the moment when the Inversion layer reach the Residual layer and the moment when the NPF occurs at the ground ? That could be great to have as well the RL height, and the estimated speed of the aerosol.

L220 : So you found 8 cases out of ? That would be nice to see a table showing the number of days of observations, the number of events at the ground, the number of event linked to roll vortices, the number of event linked to the RL, and the number of event that are not yet related to anything. And precise the type of events (classic banana or burst of particles at higher diameter than 3nm ? Again here you said these cases were not observed at the same time : Could you provide a table with their main characteristic : Start time, duration, GR, diameter at time start ?

L236 : please replace transported event by " transported event"

L237 : They occurred when the conditions inside the ML were less favourable for nucleation ⇒ could you please explain what you have in mind ?

L 246- 252 : could you provide the number and the percentage ?

Reference that might be added to your manuscript : A lot of work have been done by the French group of the LaMP to study NPF events on the ground at an altitude site but also using aircraft measurements. You should cite some of them in your paper… Aircraft observations for links between altitude and NPF : Crumeyrolle et al 2010, Altitude site : Boulon, et al.: Investigation of nucleation events vertical extent: a long term study at two different altitude sites, Atmos. Chem. Phys., 11, 5625–5639, https://doi.org/10.5194/acp-11-5625-2011, 2011. C. Rose, et al. , Frequent nucleation events at the high altitude station of Chacaltaya (5240 m a.s.l.), Bolivia,https://doi.org/10.1016/j.atmosenv.2014.11.015. H Venzac, et al - 2007 - Aerosol and ion number size distributions were measured at the top of the Puy de Dôme (1465 m above the sea level) for a three-month period. The goals were to investigate the vertical extent of nucleation in the atmosphere and the effect of clouds on nucleation. J. Boulon, et al. New particle formation and ultra- fine charged aerosol climatology at a high altitude site in the Alps (Jungfraujoch, 3580 m a.s.l., Switzer- land). Atmospheric Chemistry and Physics, European Geosciences Union, 2010, 10 (19), pp.9333-9349.

Also maybe look at that one : https://www.mdpi.com/2072-4292/12/4/648. It does also look at the impact of the dynamics on the nucleation events with a clear focus on the dynamics. You can actually see that the perturbation induced by flows at different altitude might also enhanced the possibility to observed NPF events. The turbulent fluxes occurring at each layer top is inducing favourable conditions to generate NPF events.

---

## Author Comment (AC1) · 10 Dec 2020

We thank the Referee for the comments. Please see our answers below.

Comment: A number of previous studies, i.e., Nilsson et al. (2001), Stratmann et al. (2003), Stanier et al. (2004); (Wehner et al., 2007), and (Platis et al., 2016) suggested that enhanced turbulent mixing, related to the growth of daytime convective boundary layer and the lift of the inversion could cause downward mixing of the particles, which had already grown in size. In addition, there have been several recent studies that point out direct evidence for NPF occurring aloft, in the interface between the shallow convection and inversion (Chen et al., 2018; Größ et al., 2018). By using turbulence statistics and the boundary layer dynamics (Meskhidze et al., 2019) and (Zimmerman

et al., 2020) quantified the frequency of the residual layer and the ground level nucleation events and assessed their contributions (relative to other sources) to the near-surface fine particle number budgets during different seasons. The authors don't seem to acknowledge many of these studies. That leaves the impression that the residual layer nucleation and the particle entrainment into the mixed layer is a novel mechanism for explaining the appearance of >10 nm-sized particles at the near-surface layer. I would encourage the authors to clearly discuss how their research builds upon these prior studies and highlight the similarities.

Answer: In order to put this study into context we added the following background to the Introduction:

"NPF has been observed in various environments and at various altitudes inside the troposphere. The majority of NPF observations come from ground-based measurements (Kerminen et al., 2018; Kulmala et al., 2004), which can be argued to represent NPF within the mixed layer (ML). Measurements from aircrafts show that NPF is also common in the upper free troposphere (FT) (e.g. Clarke and Kapustin, 2002; Takegawa et al., 2014). Entrainment of particles formed in the upper FT was identified as an important source of CCN in the tropical boundary layer (BL) (Wang et al., 2016; Williamson et al., 2019). Measurements from high-altitude research stations also demonstrate that NPF frequently takes place in the FT, in these cases NPF was often observed in BL air that was transported to the higher altitudes (Bianchi et al., 2016; Boulon et al., 2011; Rose et al., 2017; Venzac et al., 2008).

When studying the vertical distribution of NPF in the lower troposphere one has to consider the evolution and dynamics of the BL. Nilsson et al. (2001) found that the onset of turbulent mixing correlated better with the onset of NPF at ground level than with the increase in solar radiation. The authors gave several hypotheses to why this might be. One hypothesis was that NPF starts aloft, either in the RL or in the inversion capping the shallow morning ML. As the turbulent mixing starts, the newly formed particles would be transported down and observed at the ground-level.

Many observations have supported the hypothesis put forward by Nilsson et al. (2001). Größ et al. (2018), Meskhidze et al. (2019) and Stanier et al. (2004) reported positive correlation between the onset of NPF at ground level and the breakup of the morning inversion due to beginning of convective mixing. Chen et al. (2018), Platis et al. (2015) and Siebert et al. (2004) used in situ airborne measurements and observed that NPF started during the morning on the top of a shallow ML capped by a temperature inversion at a few hundred meters above ground. The particles grew to detectable nucleation mode (sub-25 nm) sizes aloft, and when the ML began to grow due to thermally-driven convection, the particles were mixed downwards and observed at the ground-level where they further continued to grow in size. Stratmann et al. (2003) observed newly formed particles inside the RL disconnected from the shallow ML or the inversion that capped it. Furthermore, Wehner et al. (2010) observed that NPF inside the RL was connected to turbulent layers. On the other hand, Junkermann and Hacker (2018) attributed their observations of elevated ultrafine particle layers at few hundred meter altitudes in the RL to flue gas emissions from stacks with subsequent chemistry taking place during air mass transport over long distances.

The hypothesis proposed by Nilsson et al. (2001) was based on observations done in Hyytiälä, Finland, which is a rural site surrounded by boreal forests and with very clean air. However, the supporting evidence comes from measurements done in more polluted environments in Central Europe and USA. Airborne measurements done over Hyytiälä have not found NPF on top of the shallow morning ML or within the bulk of the RL, instead the NPF events seem to start within the ML (Boy et al., 2004; Laakso et al., 2007; O'Dowd et al., 2009). This might be because in the more polluted environments the RL and/or the shallow ML contains high enough concentrations of precursor vapors from anthropogenic sources, so that NPF can be initiated in the morning inversion and/or within the bulk of the RL. Interestingly, though, observations from Hyytiälä using a small instrumented airplane have frequently found nucleation mode particle layers above the ML at a much higher altitude range of ∼1500-2800 m above ground and the explanation for these layers is not clear (Leino et al., 2019; Schobesberger et al.,

2013; Väänänen et al., 2016). For example Väänänen et al. (2016) found that for the 2013-2014 airborne measurement campaigns 16/36 ($\sim$44%) profiles showed a sub-25 nm particle layer above the ML at altitudes greater than 1800 m asl.

In this study we used co-located airborne and ground-based measurements to study nanoparticles over a boreal forest in Hyytiälä, Finland. We aimed to characterize the elevated nucleation mode particle layers that were a frequent observation in the previous studies. Specifically we were looking at the following questions: (1) where in terms of atmospheric layers, how often and why do these aerosol particle layers occur, and (2) how they are related to ground-based observations, and what implications this has for data interpretation."

Comment: The airplane flight profiles seem to be different between Fig. 3 and Fig. 4. Are these two different profiles? If so, please explain.

Answer: There was a mistake in the time range given in the Fig. 3 caption. The correct time range is 12:00-13:12. Furthermore we combined the May 2, 2017 case study figures into a single figure (Fig. 1).

Comment: Fig. 4 shows that the negative flux was measured at the surface starting at 9:30 am. However, according to Fig. 3, there was no significant vertical gradient between the surface and the 1000 m. Please explain the presence of strongly negative fluxes between 9:30 am and 12:30 pm. According to Fig. 4, a new 10 nm particle mode only appeared at the ground-level at âĹij12:35 pm. So, what causes negative fluxes in the morning?

Answer: The previous correction to the time range should remove the confusion here. In addition we added some text about the particle mode and the negative flux in the morning:

"At the ground level a new particle mode with lower number concentration coupled with negative particle flux also appeared at around 10:00. It may be that these particles

were also mixed down from higher altitudes, but in the absence of airplane measurements during that time, we cannot be sure."

Comment: Please include several more case studies so the reader can compare the similarities and contrast the differences. For each case study please show the normalized spectral density plots so the reader can ascertain that there was indeed a growth event following the appearance of >10 nm-sized particles at the near-surface layer.

Answer: While a particle layer was observed on multiple flights, it is rare to find cases where one can directly observe a particle layer mixing down from the airplane and link the ground-based observations to the airborne observations. Ideally the BL development should also be clear in the lidar and the soundings so that comparison can be made to the aerosol observations. We added one more case study (May 19, 2018) to the paper. The case is analyzed in the below text and Fig. 2:

"3.2 Case study: May 19, 2018

On May 19, 2018 another case of nucleation mode particles mixing down into the ML was observed. Figure 4A shows that during the airplane's ascend the lower edge of the particle layer was observed at ~1200 m asl and the top of the layer was at 2000 m asl. The N3-10 increased in the layer from ~1000 cm-3 up to ~10000 cm-3. When the airplane descended back into the ML the N3-10 was increased to around 6000 cm-3 throughout the ML, suggesting that the particle layer was mixed into the ML. The air masses arrived from a similar sector as in the May 2, 2017 case. SO2 and CO concentrations in Hyytiälä remained low when the particles were mixed down (0.05 ppb and 127 ppb for SO2 and CO, respectively).

Figure 4B shows particle number size distribution measurements from the measurement airplane and from the field station. The particle layer was observed as increased number concentration in the smallest size channels of the SMPS at 9:00 before the airplane flew above the ML. Roughly 20 minutes later a similar-sized particle mode appeared in the ground-based data. For this day there were no particle flux data. The

new particle mode continued to grow larger inside the ML for several hours.

Figure 4C shows the TKE dissipation rate on May 18-19, 2018 from Hyytiälä and temperature soundings from Jokioinen. On May 18, 2018 the ML went up to 2500 m asl in Hyytiälä. The Jokioinen soundings show that at 6:00 the top of the RL was at about 1800 m asl, marked by the subsiding inversion left from the previous day's ML. The particle layer mixed down from approximately 2000 m asl."

Comment: Please include the flux values for each of the 8 cases shown in Fig. 8. Since the DMPS was running at the ground site, it would be interesting to know the detected start and the end time of the events, as well as the growth rate for different size particles.

Answer: We added a table that summarizes the cases and includes the negative particle flux peak values (picture of the table in Fig. 3). Regarding the growth rates we added the following sentence to the text:

"The mean growth rate of the appearing particle modes was 2.2 nm h-1 which is similar to 2.5 nm h-1 reported by Nieminen et al. (2014) for 3-25 nm particles during NPF events in Hyytiälä."

Comment: Fig. 8 shows 6-hour differences between the times when the mixed layer reaches the top of the residual layer. Please provide an explanation based on the full analysis of the meteorological data.

Answer: We added the following paragraph to the end of section 3.5 in order to explain these differences:

[revised manuscript text omitted]

---

## Author Comment (AC2) · 11 Dec 2020

We thank the Referee for the comments. It is suggested that the nucleation mode particle layers we observed might have originated from elevated upwind pollution sources, such as power station flue stacks.

As an example air mass back trajectories for the May 2, 2017 case study are shown. It is noted that the air masses arriving at 1800 m altitude above Hyytiälä at 12 UTC (this is where the aerosol particle layer was observed) traveled over a power station few hours prior to arriving in Hyytiälä.

We tested the emission hypothesis by checking if the entrained particle layer was associated with increased SO2/CO concentrations. We observed no increase in the pollutant concentrations during the day or when the particles mixed down (Fig. 1). Therefore we believe it is unlikely that these particles originated from the power station emissions.

We also checked the pollutant concentration for the second case study (May 19, 2018) we added to the manuscript (see our answer to Referee #1) and no increase in pollutant concentrations was observed when the particles mixed down at around 9:20 (Fig. 1).

Junkermann and Hacker (2018) explains that the flue stack emissions are usually released to altitudes below 400 m. In Finland the tallest chimneys are well below 200 m agl. The particle layers we observed from the Cessna were on average between 2300-2700 m above Hyytiälä. During daytime when the BL is mixing flue stack emissions would be mixed throughout the mixed layer and then stay in the residual layer the following night. One would not expect a distinct layer at the top of the RL to form. If the emissions were released into the residual layer during night, they would remain at roughly the same altitude due to lack of vertical transport during night and not be transported to the top of the RL. We think that in this case the better explanation is that the nanoparticles were formed aloft.

Comment: What are the ambient conditions relevant to particle formation or aging in the residual layer, temperature, humidity, water vapor concentration, wind speed and direction, potential condensation sink? Are aging nano-particles in this layer growing or shrinking (Kerminen et al, 2018 and references cited therein)?

Answer: According to Alonso-Blanco et al. (2017) conditions in the residual layer that would favor particle shrinkage are lack of sunlight during night and dilution because the air is cleaner. After sunrise the increased solar radiation at higher altitudes would not favor particle shrinkage. Also the lower temperature would not favor particle shrinkage. After sunrise increased solar radiation, low pre-existing aerosol particle surface area and cold temperatures would favor NPF. NPF would probably not be taking place during the night due to lack of solar radiation

Comment: 40 km is a wide range, see Fig. 1. Where is the GPS-location of the

measurements with respect to well known locations of major precursor molecule and/or primary nanoparti- cle emissions upwind? What is the flight pattern during ascents and descents? Can this be used to derive wind data from example from GPS when the Lidar is not sensitive enough?

Answer: The majority of flights were centered over Hyytiälä. We modified Figure 1 in the manuscript to also show the horizontal distribution of measurements (Fig. 2). Notable emission sources close to this area would be the city of Tampere ~60 km SW (population ~250000) from Hyytiälä and the Korkeakoski sawmill ~6km SE from Hyytiälä (Eerdekens et al., 2009). When we flew over Tampere the effect on particle number concentrations was always clear. Usually the >3 nm number concentrations increased to about 5000 cm-3 from the background 2000 cm-3 at couple hundred meters above the ground.

Also we noticed the 2011-2018 dataset was not restricted to this 40 km radius from Hyytiälä. So we remade the Figures 1 and 6 in the manuscript with the 40 km boundary condition. The average vertical number-size distribution in Figure 6 did not change much but the 3-10 nm bin showed slightly negative values above the ML. For the updated figure we only used the SMPS data (Fig. 3). Also for the temperature profile we only considered profiles when there was an increased (larger than 75th percentile) sub-25 nm number concentration in the RL (2000-3000 m).

The flight patterns were straight legs perpendicular to the mean wind direction while ascending or descending. So at least the wind direction can be inferred from the direction of the flight legs.

Comment: Why are the measurements in the ascend beginning at 200 m, the descend ends at 600 m agl? Teisko, âĹij 15 km (alt 158 m) from Hyytiälä (alt 178 m) would be a location for missed approaches and legal low altitude flying. O'Dowd et al (2009) presented profiles nearly to the ground from QUEST 2003.

Answer: We extended the lowest altitude bin to 200-400 m asl (Fig. 4). The descend

ended at 500 m asl during that flight.

Comment: Are there any indicators for example from the Hyytiälä meteorological tower that can be related to vertical mixing intensity? Potential parameters could be surface temperature or temperatures in the vertical profile. Glider pilots use ground based temperature measurements for a decision when to take off.

Answer: There is a 3d anemometer close to canopy, so in principle turbulence intensity above the canopy could be calculated.

Comment: Although an SMPS is onboard there is no size distribution presented for the case study. A complete size distribution would be a mandatory information for the interpretation as it carries information about the age of the particles (and potential distance to the source location).

Answer: We added the size distribution (Fig. 4)

Comment: For comparison of airborne and ground based data the same parameters, particle size distributions and not particles on the aircraft and air ions on the ground, should be used.

Answer: We added the particle size distribution from SMEAR II (Fig. 4). The downside is that the time resolution is not as good (10 min instead of 4 min)

Comment: Whether the vertical profiles within 2 1/2 hours in the early afternoon and another flight in the morning are comparable at all remains open, see the HYSPLIT trajectories above. The vertical profiles of the morning flight including air mass history and trajectory need to be included as well.

Answer: We changed the text to say: "During this flight no elevated particle layer was observed and the number concentrations were quite uniform with altitude in the different size ranges, staying below 1500 cm-3." The profiles are included in Fig. 4.

Comment: To investigate, whether the 1.5 nm particles grow into the size range of

3-20 nm and to disentangle between NPF in a sulphur rich environment and primary emissions a better size resolution is necessary (Junkermann et al, 2011a). See there and in subsequent papers size distributions with a finer resolution in the range below 10 nm. It needs a lagrangian flight pattern to confirm that airmass change is excluded, see Junkermann and Hacker (2015)

Answer: The SMPS measurements onboard (going down to 10 nm) and the ground-based measurements at the SMEAR II station (going down to 4 nm in the Fig. 4, the smallest size channel was noisy) do not show multiple nucleation modes. The gas measurements at the field station do not suggest sulphur rich environment.

One interpretation is that the particles were horizontally advected to the site in another air mass. However the particle layer was observed aloft first and then ∼15 min later at the field station coupled with a downward peak in particle flux suggesting that the particles were mixed down from aloft.

Comment: The observations in the 3-20 nm range are well in agreement with the patchy structure of particle number and size distributions from previous studies in the area (O'Dowd et al, 2009, Schobesberger et al, 2013, Väänänen, et al, 2016, Leino et al, 2019) as well as the patchwork blanket of power station plumes shown by Junkermann et al (2016). All these studies point towards a significant contribution from local emission hotspots. Chemical analysis from 20 years of particle research at Hyytiälä reveals that sulphur molecules and likely ammonia are among the key substances required for the production of nanoparticle clusters. A recent publication by Hao et al (2018) about measurements at Hyytiälä even requires particulate sulphate in the residual layer mixed downward to explain the observations on the ground.

Answer: In the studies mentioned the patchiness was observed inside the mixed layer but not above. It seems there are few sub-25 nm particles above the mixed layer in Hyytiälä, except for the top of the RL (Fig. 3).

The patchiness of nucleation mode particles can have other explanations such as variable cloud cover (Wehner et al., 2007), land features (O'Dowd et al., 2009), and organized convection like roll vortices (Lampilahti et al., 2020).

Comment: The sources of such sulfate particles in the atmosphere are well known and typically linked to burning processes (Bigg and Turvey, 1978, Ayers et al, 1979, Whitby et al, 1978). In Finland these sources are mostly located along the coastline, about 200 km or approximately 5-6 hours upwind of Hyytiälä, (Fig. 1, www.endcoal.org). Further examples for primary nanoparticle size, aging and emission rates are shown in the papers of Junkermann et al.

Transport via the residual layer is not an exclusive pattern, veering plumes from wind direction changes in the planetary boundary layer can explain as well the observations without additional aerosols in the residual layer (examples: O'Dowd et al, 2009, for Hyytiälä under conditions with snow covered ground or Laaksonen et al, 2005 (SPC, Italy) Junkermann and Hacker (2018)). In all cases 3D-meteorology is the key for analysis of these observations.

Answer: We do observe increased aerosol particle concentrations in the top parts of the RL. The case studies (May 2, 2017 and May 19, 2018) and ground-based observations from the BAECC campaign fit the idea that particles are mixing down from the top of the RL.

With such moving emission plumes we would expect to see changes in SO2/CO concentrations but for example in the case studies this was not observed.

Comment: Recent and historic literature is not always taken into account. There are not many airborne studies of nanoparticles, but they should be included.

Answer: We extended the Introduction, see the answer to Referee #1.

Comment: Time within the manuscript is mixed between UTC and Eastern European Summer Time (EEST) in figures and text

Answer: All time should be fixed to UTC now.

Comment: Fig. 5: Scales for Theta and water vapor are missing. The figure is not really sup- portive, it suggests a high mixed layer at night although the upper rim of the Lidar data reflect only the vertical range of the measurements. Significant TKE for vertical mixing is restricted only to daylight hours.

Answer: We added temperature soundings on top of the lidar data (Fig. 4). The point of showing the temperature soundings was to show where the temperature inversion was and it agrees with the mixed layer height based on TKE dissipation rate. During the night the vertical mixing reduces but the temperature inversion remains present and shows where the top of the night time residual layer was. In all the figures where we show the lidar data and the soundings we added the temperature scale to the top.

Comment: please take into account: a few hours upwind of Hyytiälä one of Finland's largest pollution source âĹij150 kg sulphur dioxide / his located, emitting both a large amount of primary particles and a mixture of substances relevant to nanoparticle formation independent on the time of the day

Answer: We added the following paragraph to the case study

"The air masses came from the Arctic Ocean over northern Scandinavia. They went over the west coast of Finland where there are known pollution sources (most notably the Vaskiluoto coal-fired power plant), however in Hyytiälä the $SO_2$ and CO levels remained low all day ($\sim$0.025 ppb and $\sim$115 ppb for $SO_2$ and CO, respectively). Even when the particles were observed at the surface no increase in pollutant concentrations was observed. Pollution released into the night time RL from elevated sources such as flue gas stacks would be expected to form layers at roughly the altitudes where the emissions occured below few hundred meters. This is because of the lack of vertical mixing. If the pollution was released during daytime into a ML, it would be uniformly mixed into the ML and stay like that in the RL during night (Junkermann and Hacker, 2018). The likely explanation for sub-10 and sub-3 nm particles at this altitude is NPF."

Comment: Fig. 9 should be discussed in terms of the annual variability of meteorology,
for example the intensity of convection under typical weather conditions in Finland. The intensity of the vertical mixing process described in the manuscript is dependent on surface conditions (snow until the end of March?) and surface and vertical profile temperatures throughout the year.

Answer: The days were sunny spring days without snow (and one sunny day in July). On such days the ML is expected to be well-mixed and the particles should reach the surface in less than an hour or so (Stull, 1988). We looked at the soundings released at $\sim$11:20 and $\sim$17:20 from Hyytiälä during these days. The approximately constant potential temperature profiles suggest a well-mixed layer (Fig. 5).

Comment: Fig. 10 is outdated and needs severe revision. Sources are not always on the ground they can be elevated as well

Answer: We will add some trees to represent biogenic emissions and smokestacks to represent anthropogenic emissions of precursors. However relative to the $\sim$2.5 km asl altitude where the particle layers were on average observed (Fig. 3) we find it does not make much difference to distinguish between sources that are at $\sim$150m altitude or at surface.

Comment: Platis et al 2015 should be Platis et al, 2016

Answer: 2015 should be correct (see: https://link.springer.com/article/10.1007/s10546-015-0084-y)

Comment: Junkermann and Hacker (2018) is cited in the text but missing in the reference list.

Answer: It is added to the list

References:

Alonso-Blanco, E., Gómez-Moreno, F. J., Núñez, L., Pujadas, M., Cusack, M. and Artíñano, B.: Aerosol particle shrinkage event phenomenology in a South Euro-

pean suburban area during 2009–2015, Atmospheric Environment, 160, 154–164, doi:10.1016/j.atmosenv.2017.04.013, 2017.

Eerdekens, G., Yassaa, N., Sinha, V., Aalto, P. P., Aufmhoff, H., Arnold, F., Fiedler, V., Kulmala, M. and Williams, J.: VOC measurements within a boreal forest during spring 2005: on the occurrence of elevated monoterpene concentrations during night time intense particle concentration events, Atmospheric Chemistry and Physics, 9(21), 8331–8350, doi:https://doi.org/10.5194/acp-9-8331-2009, 2009.

Wehner, B., Siebert, H., Stratmann, F., Tuch, T., Wiedensohler, A., Petäjä, T., Dal Maso, M. and Kulmala, M.: Horizontal homogeneity and vertical extent of new particle formation events, Tellus B, 59(3), 362–371, doi:10.1111/j.1600-0889.2007.00260.x, 2007.

O'Dowd, C. D., Yoon, Y. J., Junkermann, W., Aalto, P., Kulmala, M., Lihavainen, H. and Viisanen, Y.: Airborne measurements of nucleation mode particles II: boreal forest nucleation events, Atmos. Chem. Phys., 9(3), 937–944, doi:10.5194/acp-9-937-2009, 2009.

Lampilahti, J., Manninen, H. E., Leino, K., Väänänen, R., Manninen, A., Buenrostro Mazon, S., Nieminen, T., Leskinen, M., Enroth, J., Bister, M., Zilitinkevich, S., Kangasluoma, J., Järvinen, H., Kerminen, V.-M., Petäjä, T. and Kulmala, M.: Roll vortices induce new particle formation bursts in the planetary boundary layer, Atmospheric Chemistry and Physics, 20(20), 11841–11854, doi:https://doi.org/10.5194/acp-20-11841-2020, 2020.

Stull, R. B.: An Introduction to Boundary Layer Meteorology, Softcover reprint of the original 1st ed. 1988 edition., Springer, Dordrecht., 1988.
* * *
[Figure]

[Figure]

[Figure]

**Fig. 1.**

[Figure]

(A)     (B)

**Fig. 2.**

[Figure]

Fig. 3.

(A)

Ascend

| | |
|---|---|
| 1.5-3 nm (02:26-03:07) | |
| 3-10 nm (02:26-03:07) | |
| >10 nm (02:26-03:07) | |
| 1.5-3 nm (12:00-12:36) | |
| 3-10 nm (12:00-12:36) | |
| >10 nm (12:00-12:36) | |
| ML height | |

Descend

| | |
|---|---|
| 1.5-3 nm (03:18-03:55) | |
| 3-10 nm (03:18-03:55) | |
| >10 nm (03:18-03:55) | |
| 1.5-3 nm (12:48-13:20) | |
| 3-10 nm (12:48-13:20) | |
| >10 nm (12:48-13:20) | |

Particle number conc., [cm⁻³]

(C) Temperature, [C]

Temperature sounding

Top of the RL

TKE dissipation rate, [m²s⁻³]

Altitude, [m asl.]

Hour of the day, UTC

(B) Cessna 172

Diameter, [m]

dN/dlogDp, [cm-3]

Particle layer observed at 12:24

Hour of the day, UTC

SMEAR II station

New particle mode observed at 12:36

Particle flux, [10⁹ m⁻² s⁻¹]

dN/dlogDp, [cm-3]

Hour of the day, UTC

Fig. 4.

[Figure]

**Fig. 5.**

---

## Author Comment (AC3) · 12 Dec 2020

We thank the Referee for the comments. Our responses are below:

Comment: Roughly isokinetic sampling : Could you please be more precise. The inlet is either isokinetic with a control of the flow within the inlet or not isokinetic. It seems that you are controlling it with a valve and with a constant speed of the Cessna. Therefore most of the time the sampling should be isokinetic. However, roughly is too vague. What are the deviation from the isokinetic conditions ? This condition has a large impact on the measurement quality and therefore on their validity. Please correct and add more information about that.

Answer: From Schobesberger et al (2013) (reference in the manuscript): "The aerosol

inlet's design was adopted from the University of Hawai'i shrouded solid diffuser inlet design originally presented in McNaughton et al. (2007) for use aboard a DC-8 aircraft. Our inlet is a downsized version of it, suiting the lower cruising speed of the Cessna."

Detailed characterization of the inlet can be found in McNaughton et al. (2007). In our measurement range (<400 nm) the inlet losses should be negligible.

Inside the main sampling line the velocity of sample air was ∼2 m/s (∼47 lpm controlled by a manual valve), while the instruments (UCPC: 1.5 lpm, PSM: 2.5 lpm, SMPS: 1 or 4 lpm) drew the air at the core sampling inlets between ∼0.5-2 m/s. Under these conditions considerations of isokinetic sampling are not necessary. So we removed this part from the text.

Comment: Figure 3 analysis : "The layer had increased number concentrations of sub-20nm and sub-3nm particles." in comparison to what ? The descent profile ? I think you should clearly name the reference you are comparing these results to. Moreover, you should definitely show the profiles from the early morning flight on Figure 3. That would raise no doubts that the aerosol layer was not present before the sun rise and that could give the reader a clear reference. "at this point there were no signs of the particle layer" This is misleading. The layer didn't disappear spread into lower layers, in this case the ML. Is there a threshold for the RL height ? I believe the highest is the better due to lower temperature and cleaner air. But is there any RL height range for those events ? Could you also add the ML height in this figure ?

Answer: We added the particle number concentrations in the different size ranges at altitudes below/above and in the layer to the text.

We also added the particle number concentration vertical profiles from the early morning ascend/descend to the figure (Fig. 1).

We removed the misleading sentence and instead wrote: "The airplane entered back into the ML at 12:56 and the particle number concentration was increased throughout

the ML, suggesting that the particles in the elevated layer were mixed into the ML"

We added an estimate of the ML height based on the Doppler lidar data as dashed line to the figure.

In Figure 1C the temperature soundings from Jokioinen show how a temperature inversion at the top of the previous day's ML remains at roughly the same altitude (2000-2500 m asl.) during the night and height of this inversion indicates the height of the residual layer.

Comment: L 168-171 : The NPF starts at 12:36 but the vertical particle flux show minimum values at 10:30 et 13:00. If aerosols are coming from the residual layer (around 1700m), the process is not instantaneous right ? So the NPF should be related to the minimums of Vertical particle flux occurring at 10:30 and 11:30. Can you estimate the vertical speed of the aerosols ? Is the aerosol speed playing a role in the NPF occurrence ? I would think that yes due to the fact that slow motion aerosol would have grown to much larger sizes ? Could you run the analysis also for non event days ? Is there a vertical wind speed threshold that need to be exceeded ? Also for other NPF cases linked to RL NPF events, Can you tell us more about the vertical particle flux patterns observed before the occurrence of NPF ? Is it different for each case ?

Answer: If the particles are formed at the top of the RL, disconnected from the ML, then the intensity of mixing in the ML would have no effect on the particle formation. If the particles are entrained into the ML then more intense mixing would transport the particles to the surface quicker and vice versa. Also if the ML remains quite shallow due to weak mixing it may be that the particle layer is not mixed down and remains aloft.

Buzorius et al. (2001) observed that the vertical particle flux was mostly negative during NPF events and the authors argued that the particles were probably formed aloft and mixed down.

In RL NPF we were looking for negative peak in particle flux when the nucleation mode particles were first observed. In other words the particle flux is most negative when the particles are observed for the first time since all the particles would be above the flux measurement setup and none below. As the particles are further mixed into the ML the number concentration difference above and below the flux measurement setup decreases and the particle flux becomes less negative.

Comment: Figure 6: I'm not sure what you plotted on this figure. The color code correspond to dN/dlogDp (cm-3). So is it a total concentration or is it from a specific bin ? It must be a specific bin and most probably within the fine diameter range due to the conclusions drowned. Could you please provide the percent of NPF event linked to aerosol formation in the upper layer ? Then you could used this result to justify the 75th percentile use.

Answer: The figure shows the median and the 75th percentile aerosol particle number size distribution as a function of altitude calculated from 2011-2018 flight data. We did not inspect all flight profiles during 2011-2018 for layers. However Väänänen et al. (2014) (reference in the manuscript) reported that for 2013-2014 campaigns 16/36 (∼44%) profiles had a sub-25 nm particle layer. We added this number to the Introduction.

Comment: L52 : need to define ML

Answer: we added the following definition to the Introduction when we first mention the ML:

"Type of atmospheric boundary layer where turbulence tends to uniformly mix quantities such as aerosol particle concentrations."

Comment: L147 : In the aircraft data : not well said

Answer: We replaced it with "In the airborne measurements"

Comment: Figure 7 : Need to be more precise : - early morning of June 5th : 0 − 4h ?

Is there a reason why you choose that time to determine the Residual layer ? could you provide some stat for each cases of the delay between the moment when the Inversion layer reach the Residual layer and the moment when the NPF occurs at the ground ? That could be great to have as well the RL height, and the estimated speed of the aerosol.

Answer: We chose this sounding on Jun 5 because in the next sounding the RL was already mixed into the ML. In general we used the latest temperature profile where the top of the RL was visible. We added Table 1 that shows all this information (Fig. 2). We find this analysis is not accurate enough to estimate mixing speeds for the aerosol particles though.

Comment: L220 : So you found 8 cases out of ? That would be nice to see a table showing the number of days of observations, the number of events at the ground, the number of event linked to roll vortices, the number of event linked to the RL, and the number of event that are not yet related to anything. And precise the type of events (classic banana or burst of particles at higher diameter than 3nm ? Again here you said these cases were not observed at the same time : Could you provide a table with their main characteristic : Start time, duration, GR, diameter at time start ?

Answer: The campaign was 8 months Feb-Sep in 2014. We provide Table 1 (Fig. 2) for information on the specific cases. Since this particular analysis was to study the relationship between the mixing time of the RL top into ML and the appearance time of the nucleation mode particles. We did not think that classifying other types of NPF events would add much information.

Comment: L236 : please replace transported event by "transported event"

Answer: Fixed

Comment: L 246- 252 : could you provide the number and the percentage ?

Answer: We added these to the text

[Figure]

Comment: Reference that might be added to your manuscript : A lot of work have been done by the French group of the LaMP to study NPF events on the ground at an alti- tude site but also using aircraft measurements. You should cite some of them in your paper. Aircraft observations for links between altitude and NPF: Crumey- rolle et al 2010, Altitude site: Boulon, et al.: Investigation of nucleation events ver- tical extent: a long term study at two different altitude sites, Atmos. Chem. Phys., 11, 5625–5639, https://doi.org/10.5194/acp-11-5625-2011, 2011. C. Rose, et al., Fre- quent nucleation events at the high altitude station of Chacaltaya (5240 m a.s.l.), Bo- livia,https://doi.org/10.1016/j.atmosenv.2014.11.015. H Venzac, et al - 2007 - Aerosol and ion number size distributions were measured at the top of the Puy de Dôme (1465 m above the sea level) for a three-month period. The goals were to investigate the vertical extent of nucleation in the atmosphere and the effect of clouds on nucleation. J. Boulon, et al. New particle formation and ultra- fine charged aerosol climatology at a high altitude site in the Alps (Jungfraujoch, 3580 m a.s.l., Switzer- land). Atmospheric Chemistry and Physics, European Geosciences Union, 2010, 10 (19), pp.9333-9349.

Also maybe look at that one : https://www.mdpi.com/2072-4292/12/4/648. It does also look at the impact of the dynamics on the nucleation events with a clear focus on the dynamics. You can actually see that the perturbation induced by flows at different altitude might also enhanced the possibility to observed NPF events. The turbulent fluxes occurring at each layer top is inducing favourable conditions to generate NPF events.

Answer: We thank the Referee for these references. We added more information to the Introduction regarding previous studies (see our answer to Referee #1). We added some of these studies there.

References

McNaughton, C. S., Clarke, A. D., Howell, S. G., Pinkerton, M., Anderson, B., Thornhill, L., Hudgins, C., Winstead, E., Dibb, J. E., Scheuer, E. and Maring, H.: Results from the

DC-8 Inlet Characterization Experiment (DICE): Airborne Versus Surface Sampling of Mineral Dust and Sea Salt Aerosols, Aerosol Science and Technology, 41(2), 136–159, doi:10.1080/02786820601118406, 2007.

Buzorius, G., Rannik, Ü., Nilsson, D. and Kulmala, M.: Vertical fluxes and micrometeorology during aerosol particle formation events, Tellus B, 53(4), 394–405, doi:10.1034/j.1600-0889.2001.530406.x, 2001.
* * *
[Figure]

[Figure]

Fig. 1.

Table 1: rl_h = residual layer height during night or early morning (m asl), rl_ht = time when the rl_h was observed (time when the sounding was released, hour of the day, UTC), mode_t = nucleation mode particle mode first appears (hour of the day, UTC), mode_t1/mode_t2 = nucleation mode particle mode appearance confidence interval (hour of the day, UTC), rl_t = new mixed layer reaches the top of the residual layer (hour of the day, UTC), rl_t1/rl_t2 = new mixed layer reaches the top of the residual layer confidence interval (hour of the day, UTC), bl_h = observed maximum height of the previous day's boundary layer (m asl.), dp = mean mode diameter for the newly appeared particle mode, when they first appear (nm), gr = growth rate calculated for the newly appeared partice mode (nm h$^{-1}$), pf = the value of the negative particle flux peak ($10^9$ m$^{-2}$ s$^{-1}$).

| date | rl_ht | rl_h | mode_t1 | mode_t | mode_t2 | rl_t1 | rl_t | rl_t2 | dp | bl_h | pf | gr |
|---|---|---|---|---|---|---|---|---|---|---|---|---|
| 20140328 | 5.3 | 1100 | 8.5 | 9 | 9.5 | 5.5 | 7 | 8 | 20 | 1300 | -0.25 | 2.28 |
| 20140331 | 7.6 | 2400 | 14 | 14.5 | 15 | 12 | 13.5 | 14 | 10 | 2200 | -0.06 | 2.1 |
| 20140404 | 8.5 | 2200 | 10.5 | 11 | 11.5 | 10.5 | 11 | 11.5 | 8 | 2800 | -0.04 | 1.39 |
| 20140409 | 5.5 | 1500 | 9 | 9.25 | 9.5 | 6 | 6.5 | 7 | 8 | 1800 | -0.13 | 1.18 |
| 20140415 | 5.3 | 1600 | 14.5 | 14.25 | 15 | 12 | 13 | 14 | 11 | 1700 | -0.18 | 1.94 |
| 20140422 | 0.0 | 1800 | 12 | 12.5 | 13 | 10.5 | 11 | 11.5 | 17 | 1900 | -0.17 | 1.0 |
| 20140518 | 0.0 | 1500 | 9.5 | 10 | 10.5 | 8 | 8.5 | 9 | 13 | 1900 | -0.11 | 2.91 |
| 20140705 | 5.3 | 1500 | 11 | 11.5 | 12 | 8.5 | 9 | 10 | 12 | 1700 | -0.1 | 4.83 |

**Fig. 2.**

---

## Referee Report (RR1)

Review of the manuscript:

Aerosol particle formation in the upper residual layer by

Janne Lampilahti et al, submitted to ACP, second revision

The manuscript improved compared to the initial submission. However, unfortunately flight patterns (GPS locations) for the case studies are still not included and even the altitude graph was omitted. As well in situ data on water vapor respectively relative humidity wind speed and direction are missing.

The vertical advection proved to be highly relevant for ground based nucleation mode particle events as well as horizontally advected nanoparticles (line 369-375). The two, vertical and horizontal, advection processes of particles, produced or transported elsewhere, seem to dominate the appearance of nucleation and Aitken mode particles at the SMEAR II site in Hyytiälä. It is, however, not supported by data that the nanoparticles really are produced above Hyytiälä. They appear during the day, but this might just be due to a change in wind direction. It's definitely not the same air mass in the morning and at noon.

A particle formation in the upper residual layer, and the process of mixing residual layer air and free tropospheric air, is still a highly speculative hypothesis which could not be confirmed by any data neither within the current study, nor within any other investigation in the recent literature. As some of the main components for nucleation identified in the last 20 years, sulphur and ammonia or amine compounds, are rather emitted from similar or even the same anthropogenic sources. An external mixture plus water vapor from two individual transport processes above and below an inversion layer seems to be less likely a source for nucleation.

That particles of all sizes are transported along inversion layers in a stratified atmosphere is well known. That holds for Saharan dust as well as for nanoparticles co-emitted from major anthropogenic potential nucleation mode particle precursor sources. Decoupled from the planetary boundary layer containing low volatility VOC's, such particles are not expected to grow significantly, but changes in size distributions may occur due to changing ambient conditions like temperatures and humidity. A more detailed meteorological investigation including horizontal transport and a GPS location information (missing in the current manuscript) of the current data thus could be the key to identify the sources for nucleation mode particles or their precursors within the current data set.

Some further questions arose:

Timing:

In the original version in Fig 4 the aircraft altitude was plotted against the time between 11:30 and 14:00 Fig. 3 caption says 09:30 to 12:00 UTC.  However 11:30 and 14:00 is the EET and vertical flux is at $-0.06$ at 13:00 EET

In the new manuscript (version 5) the vertical profile is now between 12:00 and 13:20 UTC and the vertical flux  - 0.06 is now still at 13:00, but now the time is given in UTC.

Location:

In the new manuscript (5) the altitude graph of Fig. 4 is no longer available. While for a statistical summary a rough average vertical pattern would be sufficient (Fig. 2), for the case studies a more

detailed altitude profile or better a three dimensional plot with the GPS position is needed. Where has the aircraft been during the 15 and 30 min horizontal flight legs? Where and how far from Hyytiälä were the profiles flown? This information is important to understand, whether and how the vertical profiles are related to the ground based data from SMEAR II. For comparison, Väänänen et al presented several time particle hotspots 15 km north and south of Hyytiälä and a local minimum just above the SMEAR station. Thus the spatial information is important.

Data:

1) Why are there no 1.5 – 3 nm data included in Fig. 4, May 19, 2017?

2) In the May 2 ascend there are enhanced particle numbers in the PSM channel (1.5 – 3 nm) above 2200 m up to 2700 m reaching about 2200 cm$^{-3}$ at 2700 m and 3-10 nm were ~ 200 – 300 cm$^{-3}$. 20 minutes later 1.5 – 3 nm particles went down to nearly 200 cm$^{-3}$. Top of the mixed layer is below 2000 m (Panel C), HYSPLIT maximum ML altitude 1750 m. How can this behavior of the smallest particles be explained despite the still high radiation levels and further growth of the MBL until 14:00 UTC? The text claims that the PSM might have problems with the altitude, line 193. How reliable are these high elevation data? The commercial Airmodus PSM is specified only down to 900 hPa. Can this limiting weakness of the PSM be the reason for the unexpected particles at very high altitude levels?

3) Typical speed for a Cessna 172 descend is 500 ft min$^{-1}$ or ~ 150 m min$^{-1}$ resulting in a vertical resolution of 300 m. What is the reason for the rapid descend, keeping in mind that the SMPS has a time resolution limit of 2 minutes. What is the rationale for the 15 min horizontal pattern during the descend? The Cessna 172 has more endurance than the 2.5 h and 10 – 15 additional minutes would improve the vertical resolution.

4) Time resolution for the SMPS is 2 min according to the text, line 121. That should be 40 size distributions within the whole flight profile (12:00 – 13:20). However, resolution in Fig 3B is only ~6 minutes. This lower time resolution would be a severe restriction for airborne profile measurements.

5) Can the size distributions be extended with the 1.5 to 3 nm and 3-10 nm measurements as in Väänänen (2016). As figure 3 was changed to 1.5 – 3, 3 – 10 and > 10 nm these data should be available.

---

## Author Response (AR2)

We thank the Referee for the comments, see our response below.

General:

In both case studies we added the GPS location colored by number concentration and vertical profiles of temperature and water vapor concentration. Also we added the wind data from the SMEAR II mast (there were no wind measurements available from the airplane during case studies). Also the analysis was expanded to take into account the new figures, especially how it can be difficult to separate horizontal variability from vertical variability in number concentrations given our flight tracks in the case studies.

We expanded on one of the hypothesis by adding the following paragraph:

"Another possibility is that the RL and the FT contain different precursor vapors that did not initiate nucleation or particle growth on their own, however when the vapors are mixed in the interface between the two layers NPF occurs. For example on May 2, 2017 Beck et al. (in preparation) measured the composition of naturally charged ions using a mass spectrometer on board an aircraft concurrently with our measurements. It was found that during the first flight (~02:30-04:00 UTC) the chemical composition was different in the RL compared to the FT. For example highly oxygenated molecules (HOMs) as well as iodine containing compounds were present in the RL while methanosulfonic acid (MSA) and sulfuric acid were detected in the FT."

Regarding transport we added the following paragraph.

"One possible explanation for the elevated nucleation mode particle layers could be long-range transport coupled with changes in the particle number size distribution such as particle shrinkage. However, it is not clear why such process would favor the RL-FT interface. If the particle emissions were released into the ML they would likely be distributed more or less uniformly throughout the RL and not be concentrated at the top of the RL. If the transported particles subsided from the FT, we would expect to see particle layers at various altitudes in the FT on different days, and the layers would not be localized at the top of the RL. We studied the origin of the airmasses in the particle layers and found that they were mostly coming from the so-called "clean sector" in the northwest of Hyytiälä (Figure 8). During other than winter months this sector is associated with non-polluted air and NPF from natural precursors (Tunved et al., 2006)."

Timing:

There was a mistake in the caption of Figure 3 in the original version. The time range in the new version is correct.

The timing of the particle flux was correct all along.

Location:

We added horizontal flight track plots to the case studies.

Data:

1) The PSM had problems working

2) Kangasluoima et al. (2016) showed that the detection efficiency and cutoff diameter of the PSM had only slight change at 60 kPa compared to 100 kPa, suggesting that the PSM would be suitable for our airborne studies.

However we have had problems with the PSM showing erroneously large concentrations towards the end of ascends usually at altitudes above 2 km. In these cases the background number concentration measured by the PSM, when a filter is placed before the inlet, starts to increase from the normal values at some altitude and keeps increasing until the descend starts.

In the first case study the N(1.5-3) increased in the layer and decreased above the layer, but still remains elevated in the FT during rest of the ascend compared to descend. Since the N(1.5-3) is elevated in the layer but decreases right above it despite the gain in altitude (similar change is seen in SMPS/UCPC) would suggest that the increase in N(1.5-3) is not only due to the elevated background problem that the PSM sometimes shows during ascends.

3) The instruments were powered by separate batteries. The main reason for the limited flight time (~2.5 h) was the battery life. Also on this day we were coordinating our flight with another airplane. The other plane was carrying an instrument (mass spectrometer) that required some periods of stable pressure in order to produce reliable data and that is why there were periods of horizontal flight.

4) The plot contained 30 size distributions the mean time resolution was about 2.2 min (we changed this to the manuscript). The first scan in the figure was at 11:58 and the last scan at 13:03. To match the profile we changed the last scan to 13:17.

5) We think the vertical profile of number concentrations in different size ranges is enough to support the analysis.

References

Beck, L., Lampilahti, J., Junninen, H., Schobesberger, S., Manninen, A., Leino, K., Quéléver, L., Dada, L., Pullinen, I., Korhonen, F., Bianchi, F., Petäjä, T., Kulmala, M., and Duplissy, J.: Chemical characterisation of negative ions above boreal forest: From ground to free troposphere, in preparation.

Kangasluoma, J., Franchin, A., Duplissy, J., Ahonen, L., Korhonen, F., Attoui, M., Mikkilä, J., Lehtipalo, K., Vanhanen, J., Kulmala, M., and Petäjä, T.: Operation of the Airmodus A11 nano Condensation Nucleus Counter at various inlet pressures and various operation temperatures, and design of a new inlet system, 9, 2977–2988, https://doi.org/10.5194/amt-9-2977-2016, 2016.